# Regulation of nitrous oxide production in low oxygen waters off the coast of Peru

Claudia Frey[1,2,*], Hermann W. Bange[2], Eric P. Achterberg[3], Amal Jayakumar[1], Carolin R. Löscher[4], Damian L. Arévalo-Martínez[2], Elizabeth León-Palmero[5], Mingshuang Sun[2], Xin Sun[1], Ruifang C. Xie[3], Sergey Oleynik[1], Bess B. Ward[1]

[1]*Department of Geoscience, Princeton University, Princeton, Guyot Hall, 08544 Princeton, USA*

[2] *Helmholtz Centre for Ocean Research Kiel, Düsternbrooker Weg 20, 24105 Kiel, Germany*

[3] *Helmholtz Centre for Ocean Research Kiel, Wischhofstr. 1-3, 24149 Kiel, Germany*

[4] *Department of Biology, Nordcee, Danish Institute for Advanced Study, University of Southern Denmark,*

[5] *Departamento de Ecología, Facultad de Ciencias, Universidad de Granada, 18071, Granada, Spain*

*current address: Department of Environmental Science, University of Basel, Bernoullistrasse 30, 4056 Basel, Switzerland

**Keywords:** ETSP, ODZ, denitrification, nitrification, $N_2O$ production, [15]N tracer incubations

**Abstract.** Oxygen deficient zones (ODZs) are major sites of net natural nitrous oxide ($N_2O$) production and emissions. In order to understand changes in the magnitude of $N_2O$ production in response to global change, knowledge on the individual contributions of the major microbial pathways (nitrification and denitrification) to $N_2O$ production and their regulation is needed. In the ODZ in the coastal area off Peru, the sensitivity of $N_2O$ production to oxygen and organic matter was investigated using [15]N-tracer experiments in combination with qPCR and microarray analysis of total and active functional genes targeting archaeal *amoA* and *nirS* as marker genes for nitrification and denitrification, respectively. Denitrification was responsible for the highest $N_2O$ production with a mean of 8.7 nmol $L^{-1}$ $d^{-1}$ but up to $118 \pm 27.8$ nmol $L^{-1}$ $d^{-1}$ just below the oxic-anoxic interface. Highest $N_2O$ production from ammonium oxidation (AO) of $0.16 \pm 0.003$ nmol $L^{-1}$ $d^{-1}$ occurred in the upper oxycline at $O_2$ concentrations of 10 - 30 µmol $L^{-1}$ which coincided with highest archaeal *amoA* transcripts/genes. Hybrid $N_2O$ formation (i.e. $N_2O$ with one N atom from $NH_4^+$ and the other from other substrates such as $NO_2^-$) was the dominant species, comprising 70 – 85 % of total produced $N_2O$ from $NH_4^+$, regardless of the ammonium oxidation rate or $O_2$ concentrations. Oxygen responses of $N_2O$ production varied with substrate, but production and yields were generally highest below 10 µmol $L^{-1}$ $O_2$. Particulate organic matter additions increased $N_2O$ production by denitrification up to 5-fold suggesting increased $N_2O$ production during times of high particulate organic matter export. High $N_2O$ yields of 2.1% from AO were measured, but the overall contribution by AO to $N_2O$ production was still an order of magnitude lower than that of denitrification. Hence, these findings show that denitrification is the most important $N_2O$ production process in low oxygen conditions fueled by organic carbon supply, which implies a positive feedback of the total oceanic $N_2O$ sources in response to increasing oceanic deoxygenation.

## Introduction

Nitrous oxide ($N_2O$) is a potent greenhouse gas (IPCC 2013) and precursor for nitric oxide (NO) radicals, which can catalyze the destruction of ozone in the stratosphere (Crutzen 1970, Johnston 1971), and is now the single most important ozone-depleting emission (Ravishankara et al. 2009). The ocean is a significant $N_2O$ source, accounting for up to one third of all- natural emissions (IPCC 2013) and this source may increase substantially as a result of eutrophication, warming, and ocean acidification (see e.g. Capone and Hutchins 2013, Breider et al. 2019). Major sites of oceanic $N_2O$ emissions are regions with steep oxygen ($O_2$) gradients (oxycline), which are usually associated with coastal upwelling regions with high primary production at the surface. There, high microbial respiratory activity during organic matter decomposition leads to the formation of anoxic waters also called oxygen deficient zones (ODZs), in which $O_2$ may decline to functionally anoxic conditions ($O_2 <10$ nmol $kg^{-1}$, Tiano et al. 2014). The most intense ODZs are found in the eastern tropical North Pacific (ETNP), the eastern tropical South Pacific (ETSP) and the northwestern Indian Ocean (Arabian Sea). The anoxic waters are surrounded by large volumes of hypoxic waters (below 20 µmol $L^{-1}$ $O_2$) which are strong net $N_2O$ sources (Codispoti 2010; Babbin et al. 2015). Latest estimates of global, marine $N_2O$ fluxes (Buitenhuis et al. 2018, Ji et al. 2018) agree well with the 3.8 Tg N $y^{-1}$ (1.8 – 9.4 Tg N $y^{-1}$) reported by the IPCC (2013), but have large variability in the resolution on the regional scale, particularly along coasts where $N_2O$ cycling is more dynamic. The expansion of ODZs is predicted in global change scenarios and has already been documented in recent decades (Stramma et al. 2008, Schmidtko et al. 2017). This might lead to further intensification of marine $N_2O$ emissions, which will constitute a positive feedback on global warming (Battaglia and Joos, 2018). However, decreasing $N_2O$ emissions have also been predicted based on reduced nitrification rates due to reduced primary and export production (Martinez-Rey et al. 2015, Landolfi et al. 2017) and ocean acidification (Beman et al. 2011, Breider et al. 2019). The parametrization of $N_2O$ production and consumption in global ocean models is crucial for realistic future predictions, and therefore better understanding of their controlling mechanisms is needed.

$N_2O$ can be produced by both nitrification and denitrification. Nitrification is a two-step process, comprising the oxidation of ammonia ($NH_3$) to nitrite ($NO_2^-$) (ammonia oxidation, AO) and $NO_2^-$ to nitrate ($NO_3^-$) ($NO_2^-$ oxidation). The relative contributions to AO by autotrophic ammonia-oxidizing archaea (AOA) and ammonia-oxidizing bacteria (AOB) have been inferred, based on the abundance of the archaeal and bacterial *amoA* genes, which encode subunit A of the key enzyme ammonia monooxygenase (e.g. (Francis et al. 2005, Mincer et al. 2007, Santoro et al. 2010, Wuchter et al. 2006)). These studies consistently revealed the dominance of archaeal over bacterial ammonia oxidizers, particularly in marine settings (Francis et al. 2005, Wuchter et al. 2006, Newell et al. 2011). In oxic conditions, AO by AOB and AOA forms $N_2O$ as a by-product (Anderson 1964; Vajrala et al. 2013; Stein 2019) and AOA contribute significantly to $N_2O$ production in the ocean (Santoro et al. 2011; Löscher et al. 2012). While hydroxylamine ($NH_2OH$) was long thought to be the only obligate intermediate in AO, NO has recently been identified as an obligate intermediate for AOB (Caranto and Lancaster 2017) and presumably AOA (Carini et al. 2018). Both intermediates are present in and around ODZs and correlated with nitrification activity (Lutterbeck et al. 2018, Korth et al. 2019). Specific details about the precursor of NO to form $N_2O$ in AOA remains controversial. Stiegelmeier et al. (2014) concluded that NO is derived from $NO_2^-$ reduction to form $N_2O$, while Carini et al. (2018) hypothesized that NO is derived from $NH_2OH$ oxidation, which can then form $N_2O$. A hybrid $N_2O$ production mechanism in AOA has been suggested, where NO from $NO_2^-$ reacts with $NH_2OH$ from $NH_4^+$, which is thought to be abiotic, i.e., non-enzymatic (Koslovski et al. 2016). Abiotic $N_2O$ production, also known as chemodenitrification, from intermediates like $NH_2OH$, NO or $NO_2^-$ can occur under acidic conditions (Frame et

al. 2017), or in the presence of reduced metals like Fe or Mn and catalyzing surfaces (Zhu-Barker et al. 2015), but the evidence of abiotic $N_2O$ production/chemodenitrification in ODZs is still lacking.

When $O_2$ concentrations fall below 20 µmol L$^{-1}$, nitrifiers produce $N_2O$ from $NO_2^-$, a process referred to as nitrifier - denitrification (Frame & Casciotti 2010), which has been observed in cultures of AOB (Frame & Casciotti 2010) and AOA (Santoro et al. 2011). During nitrifier-denitrification (and denitrification), two $NO_2^-$ molecules form one $N_2O$, which thus differentiates this process from hybrid $N_2O$ production. It has also been suggested that high concentration of organic particles create high $NO_2^-$ and low-$O_2$ microenvironments enhancing nitrifier - denitrification (Charpentier et al. 2007). Overall, the yield of $N_2O$ per $NO_2^-$ generated from AO is lower in AOA then AOB (Hink et al. 2017a, 2017b) but it should be noted that the degree to which $N_2O$ yield increases with decreasing $O_2$ concentrations varies with cell density in cultures and among field sites (Cohen & Gordon 1978; Yoshida 1988; Goreau et al. 1980; Frame & Casciotti 2010, Santoro et al. 2011, Löscher et al. 2012, Ji et al. 2015a, 2018a).

The anaerobic oxidation of ammonia by $NO_2^-$ (anammox) to form $N_2$ is strictly anaerobic and important in the removal of fixed N from the system, but it is not known to contribute to $N_2O$ production (Kartal et al. 2007, van der Star et al. 2008, Hu et al. 2019). In suboxic and $O_2$ free environments, oxidized nitrogen is respired by bacterial denitrification, which is the stepwise reduction of $NO_3^-$ to elemental $N_2$ via $NO_2^-$, NO and $N_2O$. $N_2O$ as an intermediate can be consumed or produced, but at the core of the ODZ $N_2O$ consumption through denitrification is enhanced, leading to an under saturation in this zone (Bange 2008, Kock et al. 2016). Reducing enzymes are highly regulated by $O_2$ concentrations and of the enzymes in the denitrification sequence, $N_2O$ reductase is the most sensitive to $O_2$ (Zumft 1997), which can lead to the accumulation of $N_2O$ along the upper and lower ODZ boundaries (Kock et al. 2016). $N_2O$ accumulation during denitrification is mostly linked to $O_2$ inhibiting the $N_2O$ reductase, but other factors such as sulfide accumulation (Dalsgaard et al. 2014), pH (Blum et al. 2018), high $NO_3^-$ or $NO_2^-$ concentrations (Ji et al. 2018), or copper limitation (Granger and Ward 2003) may also be relevant. Recent studies contrast the view of nitrification vs. denitrification as the main $N_2O$ source in ODZs (Nicholls et al. 2007, Babbin et al. 2015, Ji et al. 2015a, Yang et al. 2017). They show the importance of denitrification in $N_2O$ production in the ETNP from model outputs (Babbin et al. 2015) and in the ETSP from tracer incubation experiments (Dalsgaard et al. 2012, Ji et al. 2015a), based on natural abundance isotopes in $N_2O$ (Casciotti et al. 2018) or from water mass analysis of apparent $N_2O$ production (ΔN2O) and $O_2$ utilization (AOU) (Carrasco et al. 2017).[45,46]$N_2O$ production from the addition of $^{15}N$-labeled $NH_4^+$, $NO_2^-$ and/or $NO_3^-$ revealed nitrification as a source of $N_2O$ within the oxic-anoxic interface, but overall denitrification dominated $N_2O$ production with higher rates at the interface and in anoxic waters (Ji et al. 2015a, 2018a). Denitrification is driven by organic matter exported from the photic zone and fuels blooms of denitrifiers leading to high $N_2$ production (Dalsgaard et al. 2012, Jayakumar et al. 2009, Babbin et al. 2014). Denitrification to $N_2$ is enhanced by organic matter additions and the degree of stimulation varies with quality and quantity of organic matter (Babbin et al. 2014). Because $N_2O$ is an intermediate in denitrification, we hypothesize that its production should also be stimulated by organic matter, possibly leading to episodic and variable $N_2O$ fluxes.

$N_2O$ concentration profiles around ODZs appear to be at steady state (Babbin et al. 2015), but are much more variable in regions of intense coastal upwelling where high $N_2O$ emissions can occur (Arévalo-Martínez et al. 2015). The contributions of and controls on the two $N_2O$ production pathways under different conditions of $O_2$ and organic matter supply, are not well understood and may contribute to this variability. Hence, the goal of this study is to understand the factors regulating $N_2O$ production around ODZs in order to better constrain how future

changes in $O_2$ concentration and carbon export will impact production, distribution and emissions of oceanic $N_2O$. Our goal was to determine the impact of $O_2$ and particulate organic matter on $N_2O$ production rates using $^{15}N$ tracer experiments in combination with qPCR and functional gene microarray analysis of the marker genes, *nirS* for denitrification and *amoA* for AO by archaea, to assess how the abundance and structure of the community impacts $N_2O$ production rates from the different pathways. $^{15}N$-labelled $NH_4^+$ and $NO_2^-$ was used to trace the production of single- ($^{45}N_2O$) and double- labelled ($^{46}N_2O$) $N_2O$ to investigate the importance of hybrid $N_2O$ production during AO along an $O_2$ gradient.

## 2 Materials and Methods

### 2.1 Sampling sites, sample collection and incubation experiments

Seawater was collected from 9 stations in the upwelling area off the coast of Peru in June 2017 onboard R/V Meteor (Figure 1). Water samples were collected from 10 L Niskin bottles on a rosette with a conductivity-temperature-depth profiler (CTD, seabird electronics 9plus system). In-situ $O_2$ concentrations (detection limit 2 $\mu$mol $L^{-1}$ $O_2$), temperature, pressure and salinity were recorded during each CTD cast. $NO_2^-$ and $NO_3^-$ concentrations were measured on board by standard spectrophotometric methods (Hydes et al. 2010) using a QuAAtro autoanalyzer (SEAL Analytical GmbH, Germany). $NH_4^+$ concentrations were determined fluorometrically using ortho- phthaldialdehyde according to Holmes et al. (1999). For $N_2O$, bubble-free triplicate samples were immediately sealed with butyl stoppers and aluminum crimps and fixed with 50 $\mu$L of saturated mercuric chloride ($HgCl_2$). A 10 mL He headspace was created and after an equilibration period of at least 2 hours the headspace sample was measured with a gas chromatograph equipped with an electron capture detector (GC/ECD) according to Kock et al. (2016). The detection limit for $N_2O$ concentration is 2nM $\pm$ 0.7nM. At all experimental depths nucleic acid samples were collected by filtering up to 5 L of seawater onto 0.2 $\mu$m pore size Sterivex-GP capsule filters (Millipore, Inc., Bedford, MA, USA). Immediately after collection filters were flash frozen in liquid nitrogen and kept at -80°C until extraction.

Three different experiments were carried out at coastal stations, continental slope and offshore stations. Experiments 1 and 2 aimed to investigate the influence of $O_2$ concentration along a natural and artificial $O_2$ gradient and experiment 3 targeted the impact of large particles (>50 $\mu$m) on $N_2O$ production. Serum bottles were filled from the Niskin bottles with Tygon tubing after overflowing three times to minimize $O_2$ contamination. Bottles were sealed bubble free with grey butyl rubber septa (National Scientific) and crimped with aluminum seals immediately after filling. The grey butyl rubber septa were boiled in MilliQ for 30min to degas and kept in a He atmosphere until usage. A 3 mL helium (He) headspace was created and samples from anoxic ($O_2$ < below detection) water depths were He purged for 15min. He purging removed dissolved oxygen contamination which is likely introduced during sampling and the headspace prevents possible oxygen leakage from the rubber seals (DeBrabandere et al. 2012). Natural abundance 2000 ppb $N_2O$ carrier gas (1000 $\mu$L in He) was injected to trap the produced labeled $N_2O$ and to ensure a sufficient mass for isotope analysis. For all experiments, $^{15}N$-$NO_2^-$,$^{15}N$-$NO_3^-$, and$^{15}N$-$NH_4^+$ tracer ($^{15}N/(^{14}N+^{15}N)$ = 99 atom-%) were injected into five bottles each from the same depth to a final concentration of 0.5 $\mu$mol $L^{-1}$, except for the $NO_3^-$ incubations where 2 $\mu$mol $L^{-1}$ final concentration were anticipated to obtain 10 % label of the $NO_3^-$ pool. The fraction labeled of the substrate pools was 0.76 – 0.99 for $NH_4^+$, 0.11 – 0.99 for $NO_2^-$ , 0.055 – 0.11 for $NO_3^-$. In the $^{15}N$-$NO_3^-$ treatment, $^{14}N$-$NO_2^-$ was added to trap the

label in the product pool for $NO_3^-$ reduction rates and in the $^{15}N$-$NH_4^+$ treatment, $^{14}N$-$NO_2^-$ was added to a final concentration of 0.5 µmol $L^{-1}$ to trap the label in the product pool for AO rates.

For the $O_2$ manipulation experiments, all serum bottles were He purged and after the addition of different amounts of air saturated site water a final headspace volume of 3 mL was achieved. Site water from the incubation depth was shaken and exposed to air to reach full $O_2$ saturation. Then 0, 0.2, 0.5, 2 and 5mL $O_2$ saturated seawater was added into serum bottles and to reach final measured $O_2$ concentration of $0 \pm 0.18$ µM, $0.4 \pm 0.24$ µM, $1.6 \pm 0.12$ µM, $5.2 \pm 0.96$ µM and $11.7 \pm 1.09$ µM in seawater. For the $^{15}N$-$NO_3^-$ incubations two more $O_2$ treatments with $21.5 \pm 2.8$ and $30.2 \pm 3.35$ µM $O_2$ were carried out to extend the range of a previous study in which $N_2O$ production from $^{15}NO_3^-$ did not decrease in the presence of up to 7 µM $O_2$ (Ji et al. 2018).The $O_2$ concentration was monitored with an $O_2$ sensor spot in one serum bottle per treatment using an $O_2$ probe and meter (FireSting, PyroScience, Aachen, Germany; Figure S1). The sensor spots are highly sensitive in the nanomolar range and prepared according to Larsen et al. (2016).

For the organic matter additions, concentrated particles > 50 µm from 3 different depths were collected with a Challenger stand-alone pump system (SAPS *in situ* pumps, Liu et al. 2005), autoclaved and He purged. 200µL of POC solution were added to each serum bottle before $^{15}N$-$NO_3^-$ or $^{14}N$-$NO_2^-$ tracer injection. The final particle concentrations and C/N ratios varied between 0.18 – 1.37 µM C and 8.1 – 15.4, respectively (Table 2). The concentration and C/N ratio of PON and POC of the stock solutions were analyzed by mass spectrometry using GV Isoprime mass spectrometer.

A set of five bottles was incubated per time course. One bottle was sacrificed at $t_0$, two bottles at $t_1$ and two at $t_2$ to determine a single rate. Total incubation times were adjusted to prevent bottle effects, which become significant after 20 h based on respiration rate measurements (Tiano et al. 2014). Hence, experiments lasted from 12 hours (at the shelf stations) to 24 hours (at the slope stations). Incubation was terminated by adding 0.1 mL saturated mercuric chloride ($HgCl_2$). All samples were stored at room temperature in the dark and shipped back to the lab.

### 2.2 Isotope measurement and rate determination

The total $N_2O$ in each incubation bottle was extracted with a purge-trap system according to Ji et al. (2015). Briefly, serum bottles were flushed with He for 35 min (38 ml $min^{-1}$), $N_2O$ was trapped by liquid nitrogen, $H_2O$ removed with an ethanol trap, a Nafion® trap and a $Mg(ClO_4)_2$ trap and $CO_2$ removed with an Ascarite $CO_2$-Adsorbance column and afterwards mass 44, 45, 46 and isotope ratios 45/44, 46/44 were detected with a GC-IRMS system (Delta V Plus, Thermo). Every two to three samples, a 20 mL glass vial with a known amount of $N_2O$ gas was measured to calibrate for the $N_2O$ concentration (linear correlation between $N_2O$ peak size and concentration, $r^2= 0.99$). The isotopic composition of the reference $N_2O$ was $\delta^{15}N= 1.75 \pm 0.10$ ‰ and $\delta^{18}O= 1.9 \pm 0.19$ ‰ present in $^{15}N^{14}N^{16}O$ or $^{14}N^{15}N^{16}O$ for $^{45}N_2O$ and the less abundant $^{15}N^{15}N^{16}O$ for $^{46}N_2O$. To evaluate the analyses of $^{15}N$-enriched $N_2O$ samples, internal isotope standards for $^{15}N_2O$ were prepared by mixing natural abundance $KNO_3$ of known $\delta^{15}N$ values with 99% $Na^{15}NO_3$ (Cambridge Isotope Laboratories) and converted to $N_2O$ using the denitrifier method (Sigman et al. 2001, Weigand et al. 2016). Measured and expected values were compared based on a binominal distribution of $^{15}N$ and $^{14}N$ within the $N_2O$ pool (Frame et al. 2017).

After $N_2O$ analysis, samples incubated with $^{15}NH_4^+$ and $^{15}NO_3^-$ were analyzed for $^{15}NO_2^-$ to determine rates of $NH_4^+$ oxidation and $NO_3^-$ reduction, respectively. The individual sample size, adjusted to contain 20 nmol of $N_2O$, was transferred into 20 mL glass vials and He purged for 10 min. $NO_2^-$ was converted to $N_2O$ using sodium

azide in acetic acid (McIlvin and Altabet, 2005) and the nitrogen isotope ratio was measured on a Delta V Plus (Thermo).

For each serum bottle, total $N_2O$ concentration (moles) and $^{45}N_2O/^{44}N_2O$ and $^{46}N_2O/^{44}N_2O$ ratios were converted to moles of $^{44}N_2O$, $^{45}N_2O$ and $^{46}N_2O$. $N_2O$ production rates were calculated from the slope of the increase in mass 44, 45 and 46 over time (Figure S2). To quantify the pathways for $N_2O$ production, rates were calculated based on the equations for $N_2$ production for denitrification and anammox (Thamdrup and Dalsgaard, 2002). In incubations with $^{15}NH_4^+$ and unlabeled $NO_2^-$, it is assumed that AO produces $^{46}N_2O$ from two labeled $NH_4^+$ (equation 1) and some $^{45}N_2O$-labeled $N_2O$ based on binomial distribution (equation 2). If more single labelled $N_2O$ is produced than expected (equation 2 and 3), a hybrid formation of one nitrogen atom from $NH_4^+$ and one from $NO_2^-$ (equation 4) is assumed to be taking place as found in archaeal ammonia oxidizers (Kozlowski et al. 2016). In incubations with $^{15}NO_2^-$, we assume that $^{46}N_2O$ comes from nitrifier-denitrification or denitrification, which cannot be distinguished (equation 1). Hence, any production of $^{45}N_2O$ not attributed to denitrification stems from hybrid $N_2O$ formation by archaeal nitrifiers (equation 4). In incubations with $^{15}NO_3^-$, denitrification produces $^{46}N_2O$ and was the only process considered and hence was calculated based on equation (1). Rates (R) are calculated as nmol $N_2O$ $L^{-1}$ $d^{-1}$ (Trimmer et al. 2016):

(1) $R_{external} = slope^{46}N_2O \times (f_N)^{-2}$

(2) $R_{expected} = slope^{46}N_2O \times 2 \times (1 - f_N) \times (f_N)^{-1}$

(3) $R_{above} = slope^{45}N_2O - p^{45}N_2O_{expected}$

(4) $R_{hybrid} = (f_N)^{-1} \times \left( slope^{45}N_2O + 2 \times slope^{46}N_2O \times \left(1 - f_N^{-1}\right) \right)$

(5) $R_{total} = pN_2O_{external} + pN_2O_{hybrid}$

where $f_N$ is the fraction of $^{15}N$ in the substrate pool ($NH_4^+$, $NO_2^-$ or $NO_3^-$), which is assumed to be constant over the incubation time. Hence, changing $f_N$ due to any other concurrent N-consumption or production process during the incubation is neglected. Nevertheless, the assumption of constant $f_N$ has implications that may affect the results. There is a potential for overestimating hybrid $N_2O$ production in $^{15}NO_2^-$ incubations by 5% in samples with high $NO_3^-$ reduction rates. But in incubations from anoxic depths with high $NO_3^-$ reduction rates, no hybrid $N_2O$ production was found at all. For example, accounting for a decrease in $f_N$ of the $NO_3^-$ pool by active $NO_2^-$ oxidation, the process with highest rates (Sun et al. 2017), had an effect of only $\pm$ 0.2% on the final rate estimate. The presence of DNRA complicates $^{15}N$-labelling incubations because it can change $f_N$ in all three tracer experiments. In $^{15}NO_3^-$ incubations, active DNRA produces $^{15}NO_2^-$ and $^{15}NH_4^+$ from $^{15}NO_3^-$, which can contribute to $^{46}N_2O$ production by AO. Even if a maximum DNRA rate (20 nM $d^{-1}$, Lam et al. 2009) is assumed to produce 0.02 nM $^{15}NH_4^+$ during the 24 h incubations and all of it is oxidized (maximum $N_2O$ production from AO 0.16 nM $d^{-1}$, this study) its contribution to $^{46}N_2O$ production is likely minor and within the standard error of the high $N_2O$ production rates from $NO_3$. Hence an overestimation of the $N_2O$ production rates is unlikely. The same applies in incubations with $^{15}N$-$NO_2^-$ when DNRA produces $^{15}NH_4^+$, additional $^{46}N_2O$ can be produced with a hybrid mechanism by AO. In $^{15}NO_2^-$ incubations with high starting $f_N$ (>0.7) the production of $^{14}NO_2^-$ by $NO_3^-$ reduction (which decreases $f_N$) leads to an underestimation by up to 9%, whereas in incubations with a low $f_N$ (<0.3) the effect is less (up to 3% underestimation of $N_2O$ production rates). In $^{15}NH_4^+$ incubations ($f_N$ >0.9), maximum DNRA rate would lead to an underestimation of 3.5%. Slope of $^{46}N_2O$ and slope of $^{45}N_2O$ represent the $^{46}N_2O$ and $^{45}N_2O$ production rates, which were tested for significance based on a linear regression (n=5, student t-test, $R^2$ >

0.80, p<0.05). Linear regressions that were not significantly different from zero were reported as 0. The error for each $N_2O$ production rate was calculated as the standard error of the slope. Detection limits were 0.002 nmol $L^{-1}$ $d^{-1}$ for $N_2O$ production from AO and 0.1 nmol $L^{-1}$ $d^{-1}$ for $N_2O$ production from denitrification based on the average measured standard error for rates (Dalsgaard et al. 2012). The curve-fitting tool of Sigma Plot was used for the $O_2$ sensitivity experiments. A one-way ANOVA was performed on the $N_2O$ production rates to determine if rates were significantly different between POM treatments.

The rates (R) of $NH_4^+$ oxidation to $NO_2^-$ and $NO_3^-$ reduction to $NO_2^-$ were calculated based on the slope of the linear regression of $^{15}NO_2^-$ enrichment over time (n = 5) (equation 6).

$$(6) \quad R = f_N^{-1} \times slope\delta\ ^{15}NO_2^-$$

where $f_N$ is the fraction of $^{15}N$ in the substrate pool ($NH_4^+$ or $NO_3^-$).

Yield (%) of $N_2O$ production during $NH_4^+$ oxidation was defined as the ratio of the production rates (equation 7).

$$(7) \quad Yield_{NH4} = \frac{N-N_2O\ (\frac{nM}{d})}{N-NO_2^-\ (\frac{nM}{d})} \times 100\%$$

Yields of $N_2O$ production during denitrification were calculated based on the fact that $N_2O$ is not a side product during $NO_3^-$ reduction to $NO_2^-$ but rather the next intermediate during denitrification (equation 8).

$$(8) \quad Yield_{NO3} = \frac{N-N_2O\ (\frac{nM}{d})}{N-NO_2^-\left(\frac{nM}{d}\right) + N-N_2O\ (\frac{nM}{d})} \times 100\%$$

All rates, yields and errors are reported in Table S3.

*2.3 Molecular Analysis – qPCR, Microarrays*

DNA and RNA were extracted using the DNA/RNA ALLPrep Mini Kit (Qiagen) followed by immediate cDNA Synthesis from purified and DNA-cleaned RNA using a SuperScript III First Strand Synthesis System (Invitrogen). The PicoGreen dsDNA Quantification Kit (Invitrogen) was used for DNA quantification and Quant-iT OliGreen ssDNA Quantification Kit (life technologies) was used for cDNA quantification.

The abundances of total and active *nirS* and archaeal *amoA* communities were determined by quantitative PCR (qPCR) with assays based on SYBR Green staining according to methods described previously (Jayakumar et al. 2013, Peng et al. 2013). Primers nirS1F and nirS3R (Braker et al. 1998) were used to amplify a 260-bp conserved region within the *nirS* gene. The *nirS* primers are not specific for epsilon-proteobacteria (Murdock et al. 2017), but in previous metagenomes from the ETSP epsilon-proteobacteria where below 3-4 % of the reads or not found, except in very sulfidic, coastal stations (Stewart et al. 2011, Wright et al. 2012, Ganesh et al. 2012, Schunck et al. 2013, Kavelage et al. 2015). Primers Arch-amoAF and Arch- amoAR (Francis et al. 2005) were used to quantify archaeal *amoA* abundance. A standard curve containing 6 serial dilutions of a plasmid with either an archaeal *amoA* fragment or a *nirS* fragment was used on respective assay plates. Assays were performed in a StratageneMx3000P qPCR cycler (Agilent Technologies) in triplicates of 20- 25ng DNA or cDNA, along with a no primer control and a no template control. Cycle thresholds (Ct values) were determined automatically and used to calculate the number of *nirS* or archaeal *amoA* copies in each reaction, which was then normalized to copies per milliliter of seawater (assuming 100% recovery). The detection limit was around 15 copies $mL^{-1}$ based on the Ct values of the no template control.

Microarray experiments were carried out to describe the community composition of the total and active *nirS* and archaeal *amoA* groups using the DNA and cDNA qPCR products. Pooled qPCR triplicates were purified and cleaned using the QIAquick PCR Purification Kit (Qiagen). Microarray targets were prepared according to Ward and Bouskill (2011). Briefly, dUaa was incorporated into DNA and cDNA targets during linear amplification with random octomers and a Klenow polymerse using the BioPrime kit (Invitrogen) and then labeled with Cy3, purified and quantified. Each probe is a 90-mer oligonucleotide consisting of a 70-mer archetype sequence combined with a 20-mer reference oligo as a control region bound to the glass slide. Each archetype probe represents a group of related sequences with $87 \pm 3\%$ sequence identity of the 70-mer sequence. Microarray targets were hybridized in duplicates on a microarray slide, washed and scanned using a laser scanner 4200 (Agilent Technologies) and analyzed with GenePix Pro 6.0. The resulting fluorescence ratio (FR) of each archaeal *amoA* or *nirS* probe was divided by the FR of the maximum archaeal *amoA* or *nirS* FR on the same microarray to calculate the normalized FR (nFR). nFR represents the relative abundance of each archetype and was used for further analyses.

Two different arrays were used, BCO16 which contains 99 archaeal *amoA* archetype probes representing ~8000 archaeal *amoA* sequences (Biller et al. 2012) and BCO15 which contains 167 *nirS* archetype probes representing ~2000 sequences (collected from NCBI in 2009). A total of 74 assays were performed with 21 *nirS* cDNA targets, 21 *nirS* DNA targets, 16 *amoA* cDNA targets and 16 *amoA* DNA samples. The original microarray data from BCO15 and BC016 are available via GEO (Gene Expression Omnibus; http://www.ncbi.nlm.nih.gov/geo/) at NCBI (National Center for Biotechnology Information) under GEO Accession No GSE142806.

### 2.4. Data analysis

Spearman Rank correlation was performed from all $N_2O$ production rates, AO and $NO_3^-$ reduction rates, environmental variables, *nirS* and archael *amoA* gene and transcript abundance as well as the 20 most abundant archetypes of total and active *nirS* and *amoA* using R. Only significant values ($p<0.05$) are shown. Archetype abundance (nFR) data were square-root transformed and beta-diversity was calculated with the Bray-Curtis coefficient. Alpha diversity of active and total *nirS* and *amoA* communities was estimated by calculating the Shannon diversity index using PRIMER6. Bray-Curtis dissimilarities were used to perform a Mantel test to determine significant differences between active and total communities of *nirS* and *amoA* using R (Version 3.0.2, package "vegan" (Oksanen et al., 2019). Canonical Correspondence Analysis (CCA) (Legendre & Legendre 2012) was used to visualize differences in community composition dependent upon environmental conditions using the software PAST (Hammer et al. 2001). Before CCA analysis, a forward selection (Borcard et al. 1992) of the parameters that described the environmental and biological variables likely to explain the most significant part of the changes in the archetypes was performed.

The make.lefse command in MOTHUR was used to create a linear discriminant analysis (LDA) effect size (LEfSe) (Segata et al. 2011) input file from the MOTHUR shared file. This was followed by a LEfSe (http://huttenhower.sph.harvard.edu/lefse/) to test for discriminatory archetypes between $O_2$ levels. With a normalized relative abundance matrix, LEfSe uses the Kruskal-Wallis rank sum test to detect features with significantly different abundances between assigned archetypes in the different $O_2$ levels and performs an LDA to estimate the effect size of each feature. A significant alpha of 0.05 and an effect size threshold of 2 were used for all marker genes discussed in this study.

## 3. Results

### 3.1 Hydrographic conditions

The upwelling system off Peru is a hot spot for $N_2O$ emissions (Arévalo-Martínez et al. 2015) with most intense upwelling in austral winter but maximum chlorophyll during December to March (Chavez and Messié, 2009; Messié and Chavez, 2015). The sampling campaign took place during austral fall in the absence of intense upwelling or maximum chlorophyll. The focus of this study was the region close to the coast, which has highly variable $N_2O$ concentration profiles (Kock et al. 2016) and $N_2O$ emissions (Arévalo-Martínez et al. 2015). The Peru Coastal Water (PCW, temperature <19.5°C, salinity 34.9 - 35.1) and the equatorial subsurface waters (ESSW, temperature 8-12°C, salinity 34.7 – 34.9) (Pietri et al. 2013) were the dominant water masses off the Peruvian coast sampled for $N_2O$ production rate measurements (Table 1). At the southern-most transect at $15.5° – 16°S$ a meso-scale anticyclonic mode water eddy (McGillicuddy et al. 2007), which was about to detach from the coast, was detected from deepening/shoaling of the main/seasonal pycnoclines (Bange et al. 2018, Figure S3). Generally, the stations were characterized by a thick anoxic layer (254 m – 427 m) reaching to the seafloor at two shelf stations (894, 883). $NO_2^-$ concentration accumulated only up to 2 µmol $L^{-1}$ in the secondary $NO_2^-$ maximum (SNM) at the northern transect (stations 882, 883), but up to 7.19 µmol $L^{-1}$ along the southern transect (Figure 2, station 907, 912). $N_2O$ concentration profiles showed a high variability with respect to depth and $O_2$ concentrations (Figure 2). The southern transect (station 907,912) showed the lowest $N_2O$ concentrations (5 nmol $L^{-1}$) in the center of the anoxic zones. At the same time, station 912 in the center of the eddy showed highest $N_2O$ concentration with 78.9 nmol $L^{-1}$ at $[O_2]$ below detection limit in the upper part of the anoxic zone. Above the ODZ, the maximum $N_2O$ peak ranged from 57.9 – 78.9 nmol $L^{-1}$ and was found at an $O_2$ concentration range from below detection (883, 894, 892, 912) up to 67 µmol $L^{-1}$ (907). Three stations (892, 894 and 904) showed high surface $N_2O$ concentrations of 64 nmol $L^{-1}$.

### 3.2 Depth Distribution of $N_2O$ production rates and total and active nirS and amoA abundance

$N_2O$ production varied with depth and substrate (Figure 3, Table S3). In the oxycline, highest AO (34 ± 0.1 nmol $L^{-1}$ $d^{-1}$ and 35 ± 9.2 nmol $L^{-1}$ $d^{-1}$) coincided with highest $N_2O$ production from AO (0.141 ± 0.003 nmol $L^{-1}$ $d^{-1}$ and 0.159 ± 0.003 nmol $L^{-1}$ $d^{-1}$) at both stations of the northern transect, stations 883 and 882, respectively (Figure 3(I)a, b). $NH_4^+$ oxidation and its $N_2O$ production decreased to zero in the ODZ. The rates of the reductive source pathways for $N_2O$ increased with depth. $N_2O$ production from $NO_2^-$ and $NO_3^-$ displayed similar patterns with highest production at or below the oxic -anoxic interface (Figure 3(II)). $N_2O$ production from $NO_2^-$ showed highest rates of 3.06 ± 1.17 nmol $L^{-1}$ $d^{-1}$ (912) and 2.37 ± 0.54 nmol $L^{-1}$ $d^{-1}$ (906) further south (Figure 3(II) m, q) compared to lower rates at northern stations, where the maximum rate was 0.71 ± 0.38 nmol $L^{-1}$ $d^{-1}$ (Figure 3(II) c, 883). A similar trend was found for $N_2O$ production from $NO_3^-$: lower maximum rates at northern stations with 2.7 ± 0.4 nmol $L^{-1}$ $d^{-1}$ (882) and 5.7 ± 2.8 nmol $L^{-1}$ $d^{-1}$ (883, Figure 3(II) b) and highest rates in southern transects with 7.2 ± 1.64 nmol $L^{-1}$ $d^{-1}$ (((Figure 3(II) l, 904) in transect 3 and up to 118.0 ± 27.8 nmol $L^{-1}$ $d^{-1}$ (Figure 3(II) p, 912) in transect 4. Generally, $N_2O$ production rates from $NO_2^-$ and $NO_3^-$ were 10 to 100-fold higher than from AO.

qPCR analysis detected lowest gene and transcript numbers of archaeal *amoA* and *nirS* in the surface mixed layer (Figure 3(I) k, l, 3(II)r, s). Highest archaeal *amoA* gene and transcript abundance was in the oxycline (1 – 40 µmol $L^{-1}$ $O_2$) with 24,500 ± 340 copies $mL^{-1}$ and 626 ± 29 copies $mL^{-1}$ at station 883 (Figure 3(I)c, d). *amoA* gene and transcript number decreased in the ODZ to 1000 – 6500 gene copies $mL^{-1}$ and 20 - 250 transcript

copies mL$^{-1}$. The profiles of *nirS* gene and transcript abundance were similar to each other (Figure 3(II) d, e) with highest abundance in the ODZ up to 1 x 10$^6$ copies mL$^{-1}$ and 2.9 x 10$^5$ copies mL$^{-1}$, respectively. Denitrifier *nirS* genes and transcripts peaked in the anoxic layer and were significantly correlated with N$_2$O production from NO$_2^-$ but not from NO$_3^-$. Archaeal *amoA* gene and transcript abundances were significantly correlated with AO and, N$_2$O production from AO (Figure S5). N$_2$O concentrations did not correlate with any of the measured variables (Figure S5).

### 3.3 Influence of O$_2$ concentration on N$_2$O production

N$_2$O production along the *in situ* O$_2$ gradient for the substrates NO$_2^-$ and NO$_3^-$ decreased exponentially with increasing O$_2$ concentrations (Figure 4b, c) while for NH$_4^+$, the N$_2$O production was highest at highest sampled O$_2$ concentration (Figure 4a). At *in situ* O$_2$ levels above 8.4 µmol L$^{-1}$ N$_2$O production decreased by 100% and 98% from NO$_3^-$ and NO$_2^-$, respectively (Figure 4b, c).

In the manipulated O$_2$ treatments from the oxic - anoxic interface (S11, S19) a unimodal response of N$_2$O production from NH$_4^+$ and NO$_2^-$ to O$_2$ is apparent (Figure 4d, e). Increasing and decreasing O$_2$ concentrations inhibited N$_2$O production from NH$_4^+$ and NO$_2^-$ with the highest N$_2$O production rate between 1.4 - 6 µmol O$_2$ L$^{-1}$. However, this response was only significant in sample S11 (Figure 4d, e). There was no significant response to O$_2$ concentration of N$_2$O production from NO$_3^-$. O$_2$ did not inhibit N$_2$O production from NO$_3^-$ up to 23 µmol L$^{-1}$ (Figure 4f).

The proportion of hybrid N$_2$O produced during AO, i.e., the formation of N$_2$O from one $^{15}$NH$_4^+$ and one N compound (excluding NH$_4^+$) such as NO$_2^-$, NH$_2$OH or NO, was consistently between 70 – 85 % across different O$_2$ concentrations for manipulated and natural O$_2$ concentrations (Figure 5a, c). Hybrid formation during N$_2$O production from NO$_2^-$ varied between 0 and 95% along the natural O$_2$ gradient (Figure 5b). In manipulated O$_2$ treatments hybrid formation from NO$_2^-$ did not change across different O$_2$ treatments but with respect to the original depth, 0% in sample S11 which originated from 145 m of station 892 or 78% in sample S19 from 120m of station 894 (Figure 5d).

Highest N$_2$O yields during AO (over 1%) occurred between 1.4 and 2 µmol O$_2$ L$^{-1}$, and decreased at both higher and lower O$_2$ concentrations (Figure 6a). However, only the increase in yield from nmol O$_2$ to 1.4 – 2 µmol L$^{-1}$ O$_2$ was significant (t-test, p<0.05) and the following decrease in yield was not (t-test, p>0.05). In the manipulated O$_2$ treatment of sample S19 (Figure 6c) the same significant pattern was observed, whereas in S11 highest yield was found at 12 µmol L$^{-1}$ O$_2$. N$_2$O yield during NO$_3^-$ reduction to NO$_2^-$ decreased to zero at 8.4 µmol L$^{-1}$ O$_2$ along the natural O$_2$ gradient (Figure 6b) while no significant response occurred in the manipulated O$_2$ treatments (Figure 6d). There, NO$_3^-$ reduction was decreasing with increasing O$_2$ but N$_2$O production was steady with increasing O$_2$ leading to high yields between 38.8 ± 9 % - 91.2 ± 47 % at 23 µmol L$^{-1}$ O$_2$.

### 3.4 Effect of large particulate organic matter on N$_2$O production

The autoclaving of the concentrated POM solution liberated NH$_4^+$ from the particles, reducing the N/C ratio of the particles compared to non-autoclaved particles (Table 2). The highest NH$_4^+$ accumulation is found in samples with the largest difference in N/C ratios between autoclaved and non-autoclaved particles (Table 2, 904-20m, 898-100 m). Addition of 0.17 – 1.37 µmol C L$^{-1}$ of autoclaved particles > 50 µm (Table 2) produced a significant increase in N$_2$O production by up to 5.2- and 4.8-fold in 10 and 7 out of 19 additions for NO$_2^-$ and NO$_3^-$ respectively (Figure 7a, b). There was no linear correlation of the origin (mixed layer depth, oxycline or anoxic

zone), the quality (N/C ratio) or the quantity of the organic matter on the magnitude of the increase. Only samples
S20 and S17 were not stimulated by particle addition and $N_2O$ production from denitrification did not significantly
differ from the control (Figure 7b).

### 3.5 Diversity and community composition of total and active nirS and amoA assemblages and its correlation with environmental parameters

nFR values from functional gene microarrays were used to describe the nitrifier and denitrifier community composition of AOA and *nirS* assemblages, respectively. nFR was averaged from duplicate microarrays, which replicated well ($R^2 = 0.89 - 0.99$). Alpha- diversities of *nirS* and archaeal *amoA* were not statistically different for total and active communities (students t-test, $p > 0.05$), but were overall lower for RNA ($3.2 \pm 0.3$) than DNA ($3.8 \pm 0.4$) (Table S1). Principle Coordinate Analysis of Bray–Curtis similarity for each probe group on the microarray indicated that the community structure of archaeal *amoA* genes was significantly different from that of archaeal *amoA* transcripts whereas community structure of *nirS* genes and transcripts did not differ significantly (Figure S4). To identify which archetypes were important in explaining differences in community structure of key nitrification and denitrification genes, we identified archetypes that accounted for more than 1% of the total fluorescence for their probe set and that were significantly different with respect to ambient $O_2$ using a lefse analysis (Table S2). Furthermore, we used CCA to test whether the community composition, or even single archetypes, could explain the $N_2O$ production rates.

The nFR distribution showed greater variability in the active (cDNA) AOA community than in the total community (DNA) among depths, stations and $O_2$ concentrations (Figure 8a, b). Archetypes over 1% made up between 76% (DNA) - 83% (cDNA) of the *amoA* assemblage and only 61% (DNA) - 68% (cDNA) of the *nirS* assemblage. The 4 most abundant AOA archetypes AOA55, AOA3, AOA21 and AOA32 made up 20% - 65% of the total and active community (Figure 8a, b). DNA of archetypes AOA55 and AOA79, both related to uncultured AOA in soils, significantly correlated with *in situ* $NH_4^+$ concentrations (Figure S5). DNA and cDNA from AOA3 and AOA83 were significantly enriched in oxic waters and AOA7, closely related with crenarchaeote SCGC AAA288-M23 isolated from station ALOHA near Hawaii (Swan et al. 2011), was significantly enriched in anoxic and hypoxic waters for DNA and cDNA respectively (Table S2). All other archetypes did not vary with $O_2$ levels. DNA of AOA 3, closely related to *Nitrosopelagicus brevis* (CN25), identified as the only archetype to be significantly correlated with $N_2O$ production and yield from AO (Figure S5).

The total and active denitrifier communities were dominated by Nir7, derived from an uncultured clone from the ODZ in the ETSP (Lam et al. 2009), and Nir7 was significantly more enriched in the active community (Figure 8c, d). DNA from ODZ depths of the eddy, S15 (907, 130 m) and S17 (912, 90 m), diverged most obviously from the rest and from each other (Figure 8c, d). Interestingly, these two samples were not divergent among the active *nirS* community (Figure 8c, d; Figure S4). DNA of Nir35, belonging to the Flavobacteriaceae derived from coastal waters of the Arabian Sea (Goréguès et al., 2004), was most abundant (12.3 %) at the eddy edge (S15) as opposed to the eddy center (S17) where nir167, representing Anammox sequences from Peru, was most abundant (12.0 %). Interestingly, Nir4 and Nir14, among the top 5 abundant archetypes, were significantly enriched in oxic water masses (Table S2). nFR signal of nir166, belonging to Scalindua, and Nir23 were among the top 5 abundant archetypes and significantly enriched in anoxic depths.

CCA is a direct gradient analysis, where the gradient in environmental variables is known a priori and the archetypes are considered to be a response to this gradient. Composition from total and active AOA community

did not differ between stations and all samples cluster close together (Figure S6a, b). S18 (912, 5 m) is a surface sample with lowest $NO_3^-$ concentration (8 µmol $L^{-1}$), highest temperature and salinity of the data set and the DNA is positively related with $O_2$ and driven by AOA55, AOA32 and AOA79. RNA of S17 (912, 90 m) clusters with AOA70. AOA55 was abundant and its distribution is driven by $O_2$ and $NH_4^+$ (Figure S5).

CCA clustered the denitrifier community DNA into one main group with a few exceptions (Figure S6 c). Two surface samples (S16, S18) clustered separate and were positively correlated with Nir4 and Nir14 and $O_2$. Two anoxic samples from the eddy core (S17) and eddy edge (S15) clustered separate with S17 being driven by 3 *nirS* archetypes – Nir54, Nir10 and Nir167 and S15 by Nir23, Nir35 and Nir133 (Figure S6 c). Total and active *nirS* community composition did not differ as a function of $O_2$. Although, composition of active and total *nirS* communities were not significantly different, the active community clustered slightly differently. For *nirS* RNA, surface and oxycline samples (S16 and S10) grouped together and were correlated positively with $O_2$, temperature and salinity, whereas the anoxic eddy samples did not differ from the rest (Figure S6d). $N_2O$ production from $NO_2^-$ significantly correlated with *nirS* gene and transcript abundance but both reductive $N_2O$ production pathways were not linked with a single dominant *nirS* archetype (Figure S5)

## 4. Discussion

Most samples originated from Peru Coastal Water (PCW) characterized by supersaturated $N_2O$ concentrations (Kock et al. 2016, Bourbonnais et al. 2017). Only the deepest sample (S1, 882 - 350m) saw the presence of a different water mass, the equatorial subsurface waters. Thus, our findings about regulation of $N_2O$ production at different stations probably apply to the region as a whole. Several studies indicate that water mass hydrography plays an important role in shaping microbial community diversity (Biller et al. 2012, Hamdan et al. 2012) and a coupling of *amoA* alpha diversity to physical conditions such as salinity, temperature and depth has been shown in coastal waters off Chile (Bertagnolli and Ulloa 2017). While salinity, temperature and depth were prominent factors in shaping the community compositions of nitrifiers and denitrifiers (Figure S6), for $N_2O$ production rates correlations with physical and chemical parameters were not consistent. On one hand, oxidative $N_2O$ production from $NH_4^+$ positively correlated with temperature, salinity, oxygen and negatively with depth and $PO_4^{3-}$ concentration. On the other hand, reductive $N_2O$ production from $NO_2^-$ positively correlated with $NH_4^+$ and $NO_2^-$ concentrations, but negatively with $NO_3^-$ concentrations (Figure S5), suggesting when $NO_3^-$ is abundant, denitrifiers are less likely to use $NO_2^-$ for $N_2O$ production during denitrification. Both oxidative (AO) and reductive ($NO_2^-$ and $NO_3^-$ reduction) N cycling processes produced $N_2O$ with differential effects of $O_2$ on them. Measured $N_2O$ production rates were always highest from $NO_3^-$, followed by $NO_2^-$ and $NH_4^+$, which is consistent with previous studies that showed denitrification as a dominant $N_2O$ source in Peruvian coastal waters harboring an ODZ (Ji et al. 2015a, Casciotti et al. 2018). A low contribution of AO to $N_2O$ production in low $O_2$ waters is in line with a previous study in this area estimating $N_2O$ production based on isotopomer measurements combined with a 3-D Reaction-Advection-Diffusion Box model (Bourbonnais et al. 2017). The low percentage that AO contributed to total $N_2O$ production was between 0.5 – 6%, with one exception in the shallowest sample S5 with 30 µmol $L^{-1}$ $O_2$ where AO contributed 86% to total $N_2O$ production. We found strong positive effects of decreasing $O_2$ concentration and increasing particulate matter concentrations on $N_2O$ production in the upper oxycline.

The occurrence of an anticyclonic mode water eddy at 16°S (transect 4, stations 912, 907) at the time of sampling was not unusual, as such eddies have been reported at a similar position (Stramma et al. 2013). High N loss, a large SNM with low $NO_3^-$ concentrations and strong $N_2O$ depletion in the core of ODZ of the eddy result

in reduced $N_2O$ inside of this kind of eddies as they age and are advected westward (Cornejo D`Ottone et al. 2016, Arévalo-Martínez et al. 2016). Our study found similar patterns with largest SNM (5.23 µM $NO_2^-$), lowest $NO_3^-$ (14 µmol $L^{-1}$) and $N_2O$ (4 nmol $L^{-1}$) concentrations in the eddy center. For the first time $N_2O$ production rates were measured in an eddy, and the rates of up to $118 \pm 27$ nmol $L^{-1}$ $d^{-1}$ are the highest $N_2O$ production rates from denitrification reported in the ETSP. Previously reported maximum rates were up to 86 nmol $L^{-1}$ $d^{-1}$ (Dalsgaard et al. 2012) based on $^{15}N$ tracer incubations. Much smaller maximum rates, 49 nmol $L^{-1}$ $d^{-1}$ (Bourbonnais et al. 2017) and 50 nmol $L^{-1}$ $d^{-1}$ (Farìas et al. 2009), were obtained using $N_2O$ isotope and isotopomer approaches, which provide time and process integrated signals. Hence, the deviation of maximum rates can be explained by 1) the different approaches and 2) the sampling of the core of the eddy. $N_2$ production measurements (from anammox and denitrification) were not performed in this study, but should be in future studies to account for potential artefacts by co-occurring $NO_3^-$ reduction processes. Here, it cannot be determined whether the eddy only stimulated $N_2O$ production but not $N_2$ production from denitrification (i.e. increasing the $N_2O/N_2$ yield) or if the eddy also increased complete denitrification to $N_2$ by 10 times compared to stations outside of the eddy. Considering that at some depths only incomplete denitrification (also known as "stop- and go" denitrification) to $N_2O$ is at work, it would not be surprising that $N_2O$ production can reach the same order of magnitude as $N_2$ production from complete denitrification. Aged eddies also show lower $N_2O$ concentration maxima at the upper oxycline (Arévalo-Martínez et al. 2016), which was not the case in this study where a young eddy was just about to detach from the coast. In fact, the eddy stations show the highest $N_2O$ peak in the upper oxycline within this data set. Eddies and their age imprint mesoscale patchiness and heterogeneity in biogeochemical cycling. It appears that young eddies close to the coast with high $N_2O$ concentrations and high $N_2O$ production rates have a great potential for high $N_2O$ emissions compared to aged eddies or waters surrounding eddies.

### 4.1 Effect of $O_2$ on reductive and oxidative $N_2O$ production

The relationship between $O_2$ concentrations and $N_2O$ production by nitrification and denitrification is very complex in ODZs. While poorly constrained, the reported $O_2$ threshold level (1.7 µmol $L^{-1}$ $O_2$) for reductive $N_2O$ production is lower (Dalsgaard et al. 2014) than the reported $O_2$ threshold level (8 µmol $L^{-1}$) for $N_2O$ consumption in the ETSP (Cornejo and Farías 2012). Nevertheless, the suboxic zone between $1 – 8$ µmol $L^{-1}$ $O_2$ carries high $N_2O$ concentrations indicating higher $N_2O$ production than consumption. In this study, we focused on this suboxic water masses above the ODZ and determined bulk kinetics of $O_2$ sensitivity in batch experiments, which reflect the metabolism of the microbial community. The effect of $O_2$ on $N_2O$ production differed between natural $O_2$ concentrations with varying communities vs. manipulated $O_2$ concentrations within a community. While $N_2O$ production from $NO_2^-$ and $NO_3^-$ decreased exponentially along the natural $O_2$ gradient, it did not always decrease for the manipulated $O_2$ treatments. Unchanged $N_2O$ production with higher $O_2$ levels in $NO_3^-$ treatments showed that at least a portion of the community can respond very differently to a sudden increase in $O_2$ than predicted from natural $O_2$ gradients with communities acclimated to a certain $O_2$ concentration. In the ETNP, this pattern has been observed before (Ji et al. 2018a) but the mechanism behind it is unknown. Different responses of $N_2O$ production rates to $O_2$ between *in situ* assemblages and incubated samples were not unexpected because different rates at different depths were likely not only due to $O_2$ differences but also other factors such as different organic matter fluxes and different amounts and types of $N_2O$ producers at different depths. In addition, sampling with Niskin bottles and purging can induce stress responses (Stewart et al. 2012) and shift the richness and structure of the microbial community from the *in situ* community (Torres-Beltran et al. 2019). The removal of other gases

like $H_2S$ during purging is another potential artefact. However, it is unlikely as measurable $H_2S$ concentrations have mostly been found at very shallow coastal stations (< 100 m deep) (Callbeck et al. 2018), not the environment of this study. On the contrary, high abundances (up to 12%) of sulfur oxidizing gamma proteobacteria, like SUP05 can be found in eddy-transported offshore waters where they actively contributed to autotrophic denitrification (Calbeck et al. 2018). In this study, we cannot differentiate between autotrophic or organotrophic denitrification, but a contribution of autotrophic denitrification in the eddy center is likely. Off the Chilean coast, active $N_2O$ production by denitrification was found at up to 50 µmol $L^{-1}$ $O_2$ (Farías et al. 2009). These results reinforce prior studies showing that distinct steps of multistep metabolic pathways, such as denitrification, can differ in $O_2$ sensitivity (Dalsgaard et al. 2014, Bristow et al. 2016a, 2016b). In various bacterial strains and natural communities, the $NO_3^-$ reductase enzyme (*Nar*) which catalyzes the first step in denitrification, is reportedly the most $O_2$ tolerant, followed by the more $O_2$ sensitive steps of $NO_2^-$ reduction (*Nir*) and $N_2O$ reduction (Körner und Zumft 1989, McKenney et al. 1994, Kalvelage et al. 2011). The fact that $N_2O$ production is insensitive to manipulated $O_2$ in the $NO_3^-$ treatments and not in the $NO_2^-$ treatments is evidence that it is not due to inhibition of the reduction of $N_2O$ to $N_2$ at higher $O_2$ because then both treatments would look similar. It further indicates that high $N_2O$ production from $NO_3^-$ in high oxygen treatments is unlikely an effect of anoxic microniches. While anoxic microniches in batch incubations can never be fully ruled out, there is no reason why they should systematically change $N_2O$ production in $NO_3^-$ from $NO_2^-$ incubations at the same oxygen treatment. We suggest a stimulation of incomplete denitrification, which leads to the accumulation of $N_2O$ in the serum bottles rather than a stimulation of overall denitrification rates to $N_2$. While $NO_3^-$ reduction was inhibited by higher $O_2$ concentrations, $N_2O$ production was not, leading to very high yields of $N_2O$ production per $NO_2^-$ produced. We hypothesize that there is a direct channeling of reduced $NO_3^-$ to $N_2O$ without exchange of an internal $NO_2^-$ pool with the surrounding $NO_2^-$. Long turnover times for $NO_2^-$ have been inferred from $\delta^{18}O$ of $NO_2^-$, which was fully equilibrated with water in the offshore waters (Bourbonnais et al. 2015) and more dynamic in the coastal waters (Hu et al. 2016) supporting our hypothesis. If $NO_2^-$ does not exchange, our rate estimates for $NO_3^-$ reduction based on produced $^{15}N$-$NO_2^-$ are underestimated resulting in high yields. A low $NO_2^-$ exchange rate has been shown before (Ji et al. 2018b). Based on the assumption that all labelled $N_2O$ from $^{15}NO_3^-$ has gone through the $NO_2^-$ pool, we include the $NO_2^-$ pool into calculating $f_N$. In $^{15}NO_3^-$ incubations the enrichment of the substrate pool was low ($f_N = 0.05 – 0.1$) and including $NO_2^-$ resulted in an underestimation of no more than 5 % depending on the *in situ* $NO_2^-$ concentration, and thus does not explain the high rates.

One $N_2O$ producing process not considered in this study is fungal denitrification, but it deserves mentioning because in soils and coastal sediments it contributes substantially to $N_2O$ production (Wankel et al. 2017, Shoun et al. 2012). With $^{15}N$-labelling experiments it is not possible to distinguish between bacterial and fungal denitrification. In ODZs, marine fungal communities show a wide diversity (Jebaraj et al. 2012) and a high adaptive capability is suggested (Richards et al. 2012). Most fungal denitrifiers lack the capability to reduce $N_2O$ to $N_2$, hence all $NO_3^-$ reduction results in $N_2O$ production (Richards et al. 2012). In a culture study, the fungus, *Fusarium oxysporum,* needed $O_2$ exposure before it started to denitrify (Zhou et al. 2001). To what extent marine fungi play a role in denitrification in open ocean ODZs and their $O_2$ sensitivity remains to be investigated.

$N_2O$ production from $NH_4^+$ did not decrease exponentially with increasing $O_2$ as shown previously for the ETSP (Qin et al. 2017, Ji et al. 2018a, Santoro et al. 2011). $N_2O$ production rather increased with increasing *in situ* oxygen and had an optimum between 1.4 – 6 µmol $O_2$ $L^{-1}$ in manipulated $O_2$ treatments. A similar optimum curve was observed in cultures of the marine AOA *Nitrosopumilus maritimus,* where $N_2O$ production reached

maxima at $O_2$ concentrations between 2 - 10 µmol $L^{-1}$ (Hink et al. 2017a). Furthermore, $N_2O$ production by *N. viennensis* and *N. maritimus* was not affected by $O_2$ but instead by the rate of AO (Stieglmeier et al. 2014, Hink et al. 2017a). To find out if this is the case in our study, we plotted AO rate against $N_2O$ production from $NH_4^+$ for natural and manipulated $O_2$ samples (Figure S7). The resulting significant linear fit ($R^2 = 0.75$, p<0.0001) implies that the rate of AO was the main driver for the intensity of $N_2O$ production from $NH_4^+$ and oxygen had a secondary effect.

Discrepancies in estimates of the $O_2$ sensitivity of $N_2O$ production by nitrification and denitrification are likely due to a combination of taxonomic variation as well as differences in sensitivity among the various enzymes of each pathway.

### 4.2 $N_2O$ yields and hybrid $N_2O$ formation from $NH_4^+$

$N_2O$ yields of AO were 0.15 – 2.07 % ($N_2$O-N mol/ $NO_2^-$-N mol = 1.5 x $10^{-3}$ – 20.7 x $10^{-3}$) which are at the higher end of most marine AOA culture or field studies (Hink et al 2017b, Qin et al. 2017, Santoro et al. 2011, Stieglmeier et al. 2014). Only in 2015 off the coast of Peru a higher maximum yield of 3.14% was reported (Ji et al. 2018a). While high $N_2O$ yields are usually found in low $O_2$ waters (<6 µmol $L^{-1}$), in this study AO had also high yields at higher oxygen concentrations, 0.9 % at 30 µmol $L^{-1}$ $O_2$ compared to previous studies (0.06% at > 50 µmol $L^{-1}$ Ji et al. 2018a).. In near coastal regions, higher $N_2O$ yield at higher $O_2$ concentrations expands the overall water volume where $N_2O$ production by AO contributes to high $N_2O$ concentration, which is more likely to be emitted to the atmosphere.

Insights into the production mechanism of $N_2O$ is gained from hybrid-$N_2O$ formation based on differentiating between production of single ($^{45}N_2O$) and double ($^{46}N_2O$) - labelled $N_2O$. If the production of $^{45}N_2O$ is higher than what is expected based on the binomial distribution, then an additional source of $^{14}N$ can be assumed. In $^{15}NH_4^+$ incubations, as potential $^{14}N$ substrates (besides $NH_4^+$), $NO_2^-$, $NH_2OH$ and HNO are most likely. Even though, *in situ* $NH_4^+$ is below detection in almost all water depths ($f_N > 0.9$), there remains the potential for $^{15}NH_4^+$ pool dilution by remineralization and DNRA during the incubation. Studies have shown fast turnover for $NH_4^+$, despite low $NH_4^+$ concentrations (e.g.. Klawonn et al. 2019). Even if hybrid $N_2O$ production rates are overestimated, it remains the major $N_2O$ production mechanisms from AO in this study. In future $^{15}N$ -labelling studies, co-occurrence of $NH_4^+$ production by DNRA or degradation should be measured along with $N_2O$ production to account for pool dilution. Whether hybrid $N_2O$ formation is purely abiotic, a mix of biotic and abiotic or biotic reactions, is debatable (Stieglmeier et al. 2014, Kozlowski et al. 2016, Carini et al. 2018, Lancaster et al. 2018, Stein 2019). Hybrid $N_2O$ production from $NO_2^-$ was variable with depth and oxygen, which can be explained by the different proportions of nitrifier versus denitrifier $NO_2^-$ reduction to $N_2O$. For example, in the interface sample S19 (892, 144 m, 3.69 µmol $L^{-1}$ $NO_2^-$) $N_2O$ production from $NO_2^-$ (0.72 ± 0.19 nmol $L^{-1}$ $d^{-1}$) was 20 times higher than from $NH_4^+$ (0.033 ± 0.0004 nmol $L^{-1}d^{-1}$) and no hybrid $N_2O$ formation from $NO_2^-$ was found (Figure 5d). There, the major $N_2O$ production mechanism seems to be by denitrification rather than nitrification, and even if there was a hybrid production we were not able to detect it within the given error ranges. Hybrid $N_2O$ production from $NH_4^+$ was independent of the rate at which $N_2O$ production took place and independent of the $O_2$ concentration and varied little (70 – 86% of total $N_2O$ production) during AO. Therefore, a purely abiotic reaction outside and without the vicinity of the cell can be excluded because concentrations of potential substrates for abiotic $N_2O$ production like Fe(II), Mn, NO, $NH_2OH$ vary with depth and $O_2$ concentration (Zhu-Barker et al. 2015, Kondo and Moffet 2015, Lutterbeck et al. 2018, Korth et al. 2019). Additionally, at four depths the potential

for abiotic $N_2O$ production in $^{15}NO_2^-$ addition experiments showed variations with depth and no significant impact of $HgCl_2$ fixation (Figure S9). Hence, any $^{14}N$ which is integrated into $N_2O$ to produce a hybrid/single labelled $N_2O$ has to be passively or actively taken up by the cell first (Figure 9). There, it reacts with an intermediate product ($^{15}NO$ or $^{15}NH_2OH$) of AO inside the cell. With this set of experiments, it is not possible to disentangle if hybrid production is based on an enzymatic reaction or an abiotic reaction inside the cell. Caranto et al. (2017) showed that the main substrate of $NH_2OH$ oxidation is NO, making NO an obligate intermediate of AO in AOB and suggested the existence of an unknown enzyme that catalyzes NO oxidation to $NO_2^-$ (further details also in Stein 2019). If NO is an obligate intermediate of AO in AOA (Lancaster et al. 2018), a constant rate of spontaneous abiotic or enzymatic $N_2O$ production is very likely, which always depends on the amount of NO produced in the first place. This could explain why we consistently find ~80% hybrid formation at high as well as at low AO rates. Further studies are needed to investigate the full mechanisms.

### 4.3 Effect of particulate organic matter on $N_2O$ production

A positive stimulation of $N_2O$ production from denitrification by particulate organic matter was found, indicating carbon limitation of denitrification in the ETSP. The experimental POM amendments simulated a low POC export flux and represented a flux that happens over 2 - 15 days, assuming an export flux of 3.8 mmol m$^{-2}$ d$^{-1}$ and that 8% of the total POC pool is >50 µm (Boyd et al. 1999, Martin et al. 1987, Haskell et al. 2015). We are aware that the POM collected by *in situ* pumps is a mix of suspended and sinking particles and hence the flux should be considered a rough estimate. However, the particle size (>50 µm) used in the experiments is indictive of sinking particles. The stimulation of $N_2$ production from denitrification by particulate organic matter has been shown in ODZs before (Ward et al. 2008, Chang et al. 2014), with quantity and quality of organic matter influencing the degree of stimulation (Babbin et al. 2014). In this study, amendments of POM at different degradation stages resulted in variable magnitudes of $N_2O$ production from $NO_2^-$ and $NO_3^-$ with no significant correlations between magnitude of the rates and amount, origin or quality of POM added. The processing of the particles has reduced the original N/C ratios of POM from the mixed layer more than of the POM from the ODZ, resulting in similar N/C ratios of particles from different depths. This could be one possible explanation for a lack of correlation of $N_2O$ production with origin of the POM. Furthermore, $N_2$ production was not quantified and hence it is not possible to evaluate potential relationships between overall N loss and POM additions or whether the partitioning between $N_2O$ and $N_2$ varied among treatments and depths. $N_2O/N_2$ production ratio can vary from 0 - 100% (Dalsgaard et al. 2014, Bonaglia et al. 2016). A temporary accumulation of $N_2O$ before further reduction to $N_2$ in the incubations can be ruled out as $N_2O$ accumulated linearly over time. The only station, where POM additions did not stimulate $N_2O$ production was in the center of the young eddy (912-S17). There, the highest rates of $N_2O$ production from $NO_3^-$ (118 nmol L$^{-1}$ d$^{-1}$) were found, indicating that denitrification was not carbon limited. This is consistent with previous studies on anti-cyclonic eddies, which have shown high N loss in the core of a young eddy that weakened with aging of the eddy (Stramma et al. 2013, Bourbonnais et al. 2015, Löscher et al. 2016). A direct link between the freshly produced POM fueling N loss on one hand, and decreased N loss with aging due to POM export out of the eddy on the other hand, was proposed (Bourbonnais et al. 2015, Löscher et al. 2016). In this study, the young eddy is a hot spot for $N_2O$ production.

Besides carbon availability as electron donor for denitrification, copper limitation and high $NO_3^-$ availability may play a role. Copper limitation has been argued to lead to $N_2O$ accumulation by inhibiting the copper-dependent $N_2O$ reductase (Granger and Ward 2003, Bonaglia et al. 2016), but it was not a limiting factor

640 for denitrification in the three major ODZs previously (Ward et al. 2008). Water sampling from Niskin bottles in our study was not trace metal clean and could be contaminated with Copper from the sampling system, making a limitation of trace metals in our incubations unlikely. However, OM fueled $N_2O$ production may have become limited by the availability of copper during the incubation.

   High $NO_3^-$ availability increases $N_2O$ production from denitrification in salt marshes (Ji et al. 2015b) and

645 in soils (Weier et al. 1993), systems which are generally not carbon limited. Also, at the oxic - anoxic interface of Chesapeake Bay, the ratio of $NO_2^-$ to $NO_3^-$ concentration was identified as a driver for high $N_2O$ production from $NO_3^-$ (Ji et al. 2018b). This study also found higher $N_2O$ production rates from $NO_3^-$ than $NO_2^-$, which linearly correlated with the ratio of $NO_2^-/NO_3^-$ concentrations (Figure S8). Intracellularly produced $NO_2^-$ does not seem to exchange with the surrounding pool, but ambient $NO_3^-$ is directly converted to $N_2O$, a process identified as "$NO_2^-$

650 shunting" in $N_2$ production studies (de Brabandere et al. 2014, Chang et al. 2014). POM as electron donor is an important regulator for reductive $N_2O$ production.

**4.4 Effect of abundance of total and active community composition on $N_2O$ production rates**

   The abundances of both *amoA* and *nirS* genes found in the ETSP are similar to those reported in earlier

655 studies in the ETSP (Peng et al. 2013, Ji et al. 2015a, Jayakumar et al. 2013). The *amoA* gene abundances were similar to those reported for the coastal ETSP by Lam et al. (2009), but *nirS* abundances reported here were higher than the *nirS* abundances in that study, probably due to the use of different PCR primers. The community composition of AOA did not significantly differ along the $O_2$ gradient as shown previously (Peng et al. 2013), but a significant correlation between archaeal *amoA* transcript abundance and $N_2O$ production was shown in this study.

660 The combination of qPCR and microarray analysis offered a great advantage to relate the total abundances to the production rates and additionally link particular community components to biogeochemical activities. To determine whether a particular archetype drives the correlation of $N_2O$ production by AO, a Bray-Curtis dissimilarity matrix revealed archetype AOA3 related to *Nitrosopelagicus brevis* (CN25) to be significantly correlated with the $N_2O$ production by AO. This clade is abundant in the surface ocean and typically found in high

665 abundances in the lower euphotic zone (Santoro et al. 2011, 2015). With the demonstration of high abundances of AOA3 coincident with high nitrification rates and high $N_2O$ production rates, we suggest that *Nitrosopelagicus brevis* related AOA likely play an important role in $N_2O$ production in near surface waters in the Eastern Tropical South Pacific.

   The lack of significant correlation between community composition or single members of the community

670 and reductive $N_2O$ production is consistent with the fact that *nirS* is not the enzyme directly synthesizing $N_2O$ and *nirS* communities are sources as well as sinks for $N_2O$. Taxonomic analysis of the *nirS* gene and transcripts suggested that there is high taxonomic diversity among the denitrifiers, which is likely linked to a high variability of the total denitrification gene assembly (including *nos*, *nor*, *nir*). In particular the abundance and diversity of nitric oxide reductase (*nor*), the enzyme directly synthesizing $N_2O$, would be of interest, but it is present in nitrifiers

675 and denitrifiers (Casciotti and Ward 2005) and one goal of this study was to differentiate among $N_2O$ produced by nitrifiers and denitrifers. However, *nirS* gene and transcript abundance correlated with $N_2O$ production from $NO_2^-$ making it a possible indicator for one part of reductive $N_2O$ production. It is also worth noting that anammox related *nirS* genes and transcripts (*nirS* 166, 167) contribute up to 12% of the total copy numbers putting a wrinkle on *nirS* abundance as marker gene for denitrifiers only. The subtraction of the anammox related *nirS* genes from

680 total copy numbers did not change the results from Bray-Curtis Analysis. These data indicate that the extent to

which gene or transcript abundance patterns or community composition of marker genes of processes can be used as proxies for process rate measurements is variable, likely due to complex factors, including the relative dominance of different community members, the modular nature of denitrification, differences in the level of metabolic regulation (transcriptional, translational, and enzymatic), and the range of environmental conditions being observed.

### 4.5 Summary and conclusion

In this study we used a combined approach of $^{15}$N tracer techniques and molecular techniques in order to investigate the factors that control $N_2O$ production within the upper oxycline of the ODZ in the ETSP. Our results suggest that denitrification is a major $N_2O$ source along the oxic - anoxic interface of the upper oxycline. Highest $N_2O$ production rates from $NO_2^-$ and $NO_3^-$ were found at or below the oxic-anoxic interface, whereas highest $N_2O$ production from AO was slightly shallower in the oxycline. Overall, *in situ* $O_2$ threshold below 8 µmol $L^{-1}$ favored $NO_3^-$ and $NO_2^-$ reduction to $N_2O$ and high $N_2O$ yields from AO up to 2.2%. A different pattern was observed for the community response to increasing oxygen, with highest $N_2O$ production from $NH_4^+$ and $NO_2^-$ between 1.4 – 6 µmol $L^{-1}$ $O_2$ and high $N_2O$ production from $NO_3^-$ even at $O_2$ concentrations up to 22 µmol $L^{-1}$. This study highlights the diversity of $N_2O$ production regulation and the need to conduct further experiments where single community members can be better constrained. Our experiments provide the first insights into $N_2O$ regulation by particulate organic matter in the ETSP with particles greatly enhancing $N_2O$ production (up to 5fold). Furthermore, the significant positive correlation between *Nitrosopelagicus brevis* (CN25) and $N_2O$ produced from AO could indicate its importance in $N_2O$ production and points out the great value of combining biogeochemical rate measurements with molecular analysis to investigate multifaceted $N_2O$ cycling. This study shows that short term oxygen increase can lead to high $N_2O$ production even from denitrification and extends the existing $O_2$ thresholds for high reductive $N_2O$ production up to 22 µmol $L^{-1}$ $O_2.$ Together with high $N_2O$ yields from AO up to $O_2$ levels of 30 µmol $L^{-1}$, an expansion of low oxygenated waters around ODZs predicted for the future can significantly increase marine $N_2O$ production.

Regardless of which processes are responsible for $N_2O$ production in the ODZ, high $N_2O$ production at the oxic-anoxic interface of the upper oxycline sustains high $N_2O$ concentration peaks with a potential for intense $N_2O$ emission to the atmosphere during upwelling events. An average total $N_2O$ production rate of 3.1 nmol $N_2O$ $L^{-1}$ $d^{-1}$ in a 50 m thick suboxic layer with 0 – 20 µmol $L^{-1}$ $O_2$ leads to an annual $N_2O$ efflux of 0.5 Tg N $y^{-1}$ in the Peruvian upwelling (2.22 ×$10^5$ km$^2$, Arévalo-Martínez et al. 2015), which is within the estimates based on surface $N_2O$ concentration measurements from 2012-2013 (Arévalo-Martínez et al. 2015, Bourbonnais et al. 2017). The importance of the Peru upwelling system for global $N_2O$ emissions (5 – 22% of global marine $N_2O$ emissions) is directly linked to the extreme $N_2O$ accumulations in coastal waters. Coastal $N_2O$ hotspots are well known (Bakker et al. 2014) and this study shows that they can be explained by considering denitrification as a major $N_2O$ source. While this study does not help to resolve temporal variability, manipulation experiments give valuable insights into the short-term response of $N_2O$ production to oxygen and particles. With the further parametrization of POM export as a driver for $N_2O$ production from denitrification, models may be able to better predict $N_2O$ emissions in highly productive coastal upwelling regions and to evaluate how fluxes might change with changing stratification and deoxygenation.

**Data availability:** The data presented here were archived in the SFB754 database ([www.sfb754.de](www.sfb754.de)). The $N_2O$ data are also available from the Marine Methane and Nitrous Oxide (MEMENTO) database ([https://memento.geomar.de/de/n2o](https://memento.geomar.de/de/n2o)).

**Author contributions**: CF, HWB and BW conceptualized the study. CF and MS performed experiment. CF and ELP analyzed samples. RX and EA collected POM. DLAM sampled and measured $N_2O$ concentrations. AJ performed qPCR. SO supported mass spectrometer analysis. XS supported experimental methods and assisted with data analysis. CF analyzed data and led the writing effort, with substantial contributions from all co-authors.

**Competing interests:** Authors declare no competing interests.

**Acknowledgement:** The work presented here was made possible by the DFG-funded Joint Collaborative Centre SFB754 Phase III ([http://www.sfb754.de](http://www.sfb754.de)) and by a fellowship of the German Academic Exchange Service (DAAD) program 'Postdoctoral Researchers International Mobility Experience (PRIME, ID 57350888) awarded to CF. MS was supported by the China Scholarship Council (No. 201406330054). E.L-P had a FPU-PhD fellowship (014/02917) from the Spanish Ministry of Education and a PhD International Mobility scholarship from the Universidad de Granada. CRL was funded by a EU H2020 Marie Curie Individual Fellowship (NITROX, Grant #704272) and by the Villum Foundation (Grant# 16518). EA, HWB and RX were funded by the DFG-funded Joint Collaborative Centre SFB754 program. We thank the captain and crew of R/V Meteor. Moreover, we thank the Peruvian authorities for the permission to work in their territorial waters.

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

**Tables**

**Table 1:** Overview of characteristics of samples. bd - below detection limit of Winkler method and seabird sensor (2 µmol L$^{-1}$), x - analysis includes qPCR and microarray with qPCR products, x* - only qPCR, no microarray

| ID | Stat # | Coordinates/ position | bottom depth (m) | sampling depth (m) | water column feature | Tem. (°C) | Sal. | O$_2$ (µM) seabird | NO3- (µM) | NO2- (µM) | NH4+ (µM) | $^{15}$N incubation | Tracer added | nirS | amoA |
|---|---|---|---|---|---|---|---|---|---|---|---|---|---|---|---|
| S2 | 882 | 10.95W 78.56N | 1075 | 352 | anoxic core | 11.4 | 34.82 | bd | 32.51 | 0.68 | 0.01 | depth profile | NH$_4^+$, NO$_2^-$, NO$_3^-$ | x | x |
| S1 | 882 | | 1075 | 299 | below interface | 12.1 | 34.86 | bd | 30.21 | 0.52 | 0.00 | depth profile | NH$_4^+$, NO$_2^-$, NO$_3^-$ | x | x |
| S3 | 882 | | 1075 | 259 | oxic-anoxic interface | 13.0 | 34.92 | bd | 29.39 | 1.63 | 0.01 | depth profile | NH$_4^+$, NO$_2^-$, NO$_3^-$ | x | x |
| S4 | 882 | | 1075 | 219 | above interface | 13.7 | 34.96 | 6.06 | 31.65 | 0.13 | 0.01 | depth profile | NH$_4^+$, NO$_2^-$, NO$_3^-$ | x | x |
| S5 | 882 | | 1075 | 74 | oxycline | 15.4 | 35.05 | 15.04 | 30.00 | 0.02 | 0.00 | depth profile | NH$_4^+$, NO$_2^-$, NO$_3^-$ | x | x |
| S6 | 883 | 10.78W 78.27N | 305 | 305 | anoxic core | 12.2 | 34.87 | bd | 27.27 | 1.72 | 0 | depth profile | NH$_4^+$, NO$_2^-$, NO$_3^-$ | x | x |
| S7 | 883 | | 305 | 268 | below interface | 12.8 | 34.91 | bd | 26.61 | 2.05 | 0 | depth profile | NH$_4^+$, NO$_2^-$, NO$_3^-$ | x | x |
| S8 | 883 | | 305 | 250 | oxic-anoxic interface | 13.1 | 34.92 | bd | 28.06 | 1.66 | 0 | depth profile | NH$_4^+$, NO$_2^-$, NO$_3^-$ | x | x |
| S9 | 883 | | 305 | 189 | above interface | 13.8 | 34.97 | bd | 30.47 | 0.00 | 0 | depth profile | NH$_4^+$, NO$_2^-$, NO$_3^-$ | x | x |
| S10 | 883 | | 305 | 28 | oxycline | 16.4 | 35.09 | 30.06 | 26.81 | 0.04 | 0 | depth profile | NH$_4^+$, NO$_2^-$, NO$_3^-$ | x | x |
| S19 | 892 | 12.41W 77.81N / | 1099 | 144 | below oxic-anoxic interface | 13.51 | 34.91 | bd | 19.01 | 3.69 | 0.13 | O$_2$ manipulation | NH$_4^+$, NO$_2^-$, NO$_3^-$ | x | x |
| S11 | 894 | 12.32W 77.62N/ | 502 | 120 | oxic-anoxic interface | 14.21 | 34.98 | bd | 28.92 | 0.01 | 0.00 | O$_2$ manipulation | NH$_4^+$, NO$_2^-$, NO$_3^-$ | x | x |
| S12 | 904 | 13.99W 76.66N | 560 | 179 | below interface | 13.46 | 34.94 | bd | 25.54 | 1.25 | 0.00 | POM addition (from 898) | NO$_2^-$, NO$_3^-$ | x | x* |
| S13 | 904 | | 560 | 124 | oxic-anoxic interface | 14.40 | 35.00 | bd | 27.57 | 0.09 | 0.00 | POM addition (from 898) | NO$_2^-$, NO$_3^-$ | x | x* |
| S14 | 906 | 14.28W 77.17N | 4761 | 149 | below interface | 13.70 | 34.96 | bd | 25.80 | 0.90 | 0.04 | POM addition (from 904) | NO$_2^-$, NO$_3^-$ | x | x* |
| S20 | 906 | | 4761 | 92 | oxic-anoxic interface | 14.50 | 35.00 | bd | 20.03 | 3.87 | 0.33 | POM addition (from 904) | NO$_2^-$, NO$_3^-$ | x | x* |
| S15 | 907 | 15.43W 75.43N | 800 | 130 | below interface | 14.21 | 34.98 | bd | 14.63 | 5.23 | 0.03 | POM addition (from 904) | NO$_2^-$, NO$_3^-$ | x | x |
| S16 | 907 | | 800 | 9.9 | surface | 17.82 | 35.13 | 208.3 | 16.09 | 0.99 | 0.16 | POM addition (from 904) | NO$_2^-$, NO$_3^-$ | x | x |
| S17 | 912 | 15.86W 76.11N | 3680 | 90 | below interface | 15.09 | 35.03 | bd | 19.38 | 2.85 | 0.03 | POM addition (from 906) | NO$_2^-$, NO$_3^-$ | x | x |
| S18 | 912 | | 3680 | 5 | surface | 18.05 | 35.18 | 206.0 | 8.31 | 0.47 | 0.12 | POM addition (from 906) | NO$_2^-$, NO$_3^-$ | x | x |
| S21 | 917 | 14.78W 78.04N/ | 4128 | 140 | Interface | 13.1 | 34.86 | bd | 17.3 | 3.9 | 0.0 | POM addition (from 906) | NO$_2^-$, NO$_3^-$ | x | x* |

**Table 2**: Quality (N/C), quantity (Addition µmol L$^{-1}$) and origin (station and depth) of added, autoclaved and non-autoclaved particulate organic matter (POM) and increase in NH$_4^+$ concentration after autoclaving.

| POM | Feature | Station | Depth (m) | Addition (µmol L$^{-1}$) | N/C of autoclaved POM | N/C of non-autoclaved POM | NH$_4^+$ (µM) after autoclaving |
|---|---|---|---|---|---|---|---|
| POM 1 | mixed layer depth | 898 | 60 | 0.55 | 0.10 | 0.15 | 0.7 |
| | | 904 | 20 | 0.17 | 0.09 | 0.17 | 1.56 |
| | | 906 | 50 | 0.48 | 0.07 | 0.11 | 0.57 |
| POM 2 | oxycline | 898 | 100 | 1.37 | 0.06 | 0.13 | 0.85 |
| | | 904 | 50 | 0.38 | 0.09 | 0.12 | 0.46 |
| | | 906 | 100 | 0.44 | 0.08 | 0.10 | 0.55 |
| POM 3 | anoxic zone | 898 | 300 | 0.43 | 0.09 | 0.10 | 0.15 |
| | | 904 | 150 | 0.19 | 0.10 | 0.10 | 0.20 |

**Figure Legends:**

**Figure 1**: Study area with the distribution of near-surface chlorophyll concentrations (monthly averaged for June 2017) from MODIS satellite obtained from the NASA Ocean Color Web site at 4-km resolution. Study site showing transect and station numbers, in the Eastern Tropical South Pacific during cruise M138.

**Figure 2:** Depth profiles of $O_2$, nutrients and $N_2O$ in the upper 400 m for all stations. Panel numbers 1 - 4) refer to the transect numbers.

**Figure 3:** (I) Profiles of AO, (a, e, I), $N_2O$ production rates from $NH_4^+$ (b, f, j), archaeal *amoA* gene (c, g, k) and transcript copy numbers $mL^{-1}$ (d, h, l). (II) Profiles of $NO_3^-$ reduction rates (a, f, k, o), $N_2O$ production rates from $NO_3^-$ (b, g, l, p) and $NO_2^-$ (c, h, m, q) and *nirS* gene (d, I, n, r) and transcript copy numbers $mL^{-1}$ (e, j, m, s). In (I) and (II), the panel numbers 1 – 4 correspond to transect numbers. Negative values on the y-axis represent shallower, oxic depths and the positive values represent deeper, anoxic depth (0 = interface). Shaded area indicates the anoxic zone. Note different scale for $N_2O$ production rates.

**Figure 4**: $O_2$ dependence of $N_2O$ production rates from $NH_4^+$ (a, d), $NO_2^-$ (b, e) and $NO_3^-$ (c, f). Upper panel (a-c) is $N_2O$ production along natural $O_2$ gradient from all stations. Figure 4 (b, c) are additionally zoomed in to oxygen concentrations below $5\mu$mol $L^{-1}$.Lower panel (d-f) is $N_2O$ production in manipulated $O_2$ experiments with water from oxic - anoxic interface from slope station 892 (S11, 0 µmol $L^{-1}$ $O_2$, 145m) and shelf station 894 (S19, 0 µmol $L^{-1}$, 120 m). Note different scale for $N_2O$ production rates from $NH_4^+$. Vertical error bars represent ± Standard error (n = 5 per time course). Horizontal error bars represent ± Standard error of measured $O_2$ over the time of incubations (n = 6).

**Figure 5**: $O_2$ dependency of hybrid $N_2O$ formation from $NH_4^+$ (a, c) and $NO_2^-$ (b ,d) along the natural $O_2$ gradient (a, b) and for the $O_2$ manipulations (c, d) from sample S11 (0 µmol $L^{-1}$ $O_2$, 145m) and S19 (894, 0 µmol $L^{-1}$, 120 m)

**Figure 6**: Yields (%) of $N_2O$ production during $NH_4^+$ oxidation (a, c) and during $NO_3^-$ reduction (b, d) along the natural $O_2$ gradient (a, b) and for the $O_2$ manipulations (c, d) from sample S11 (892, 0 µmol $L^{-1}$ $O_2$, 145m) and S19 (894, 0 µmol $L^{-1}$, 120 m). Error bars present ± SD calculated as error propagation.

**Figure 7**: Bar plots of $N_2O$ production after additions of autoclaved suspended and sinking particles >50 µm (See Table 2). POM1 = mixed layer depth, POM2 = oxycline, POM3 = ODZ. Error Bars represent ± SE of linear regression. * indicates significant difference to control rate (p < 0.05)

**Figure 8:** Stacked bar plot of community composition of AOA *amoA* archetypes (a, b) and *nirS* archetypes (c, d). Only archetypes over 1% contribution are shown. (a, c) total community composition (DNA). (b, d) active community composition (cDNA).

**Figure 9**: Scheme illustrating the possible reactions for hybrid $N_2O$ formation. The ellipse represents an AOA cell.

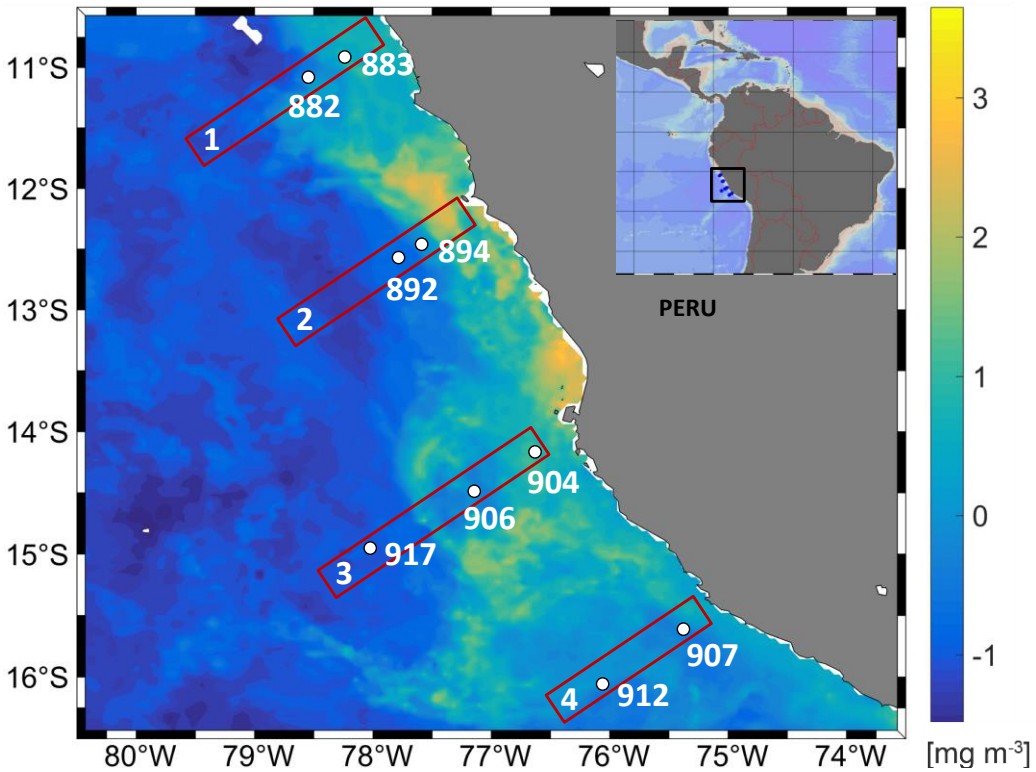

**Figure 1**

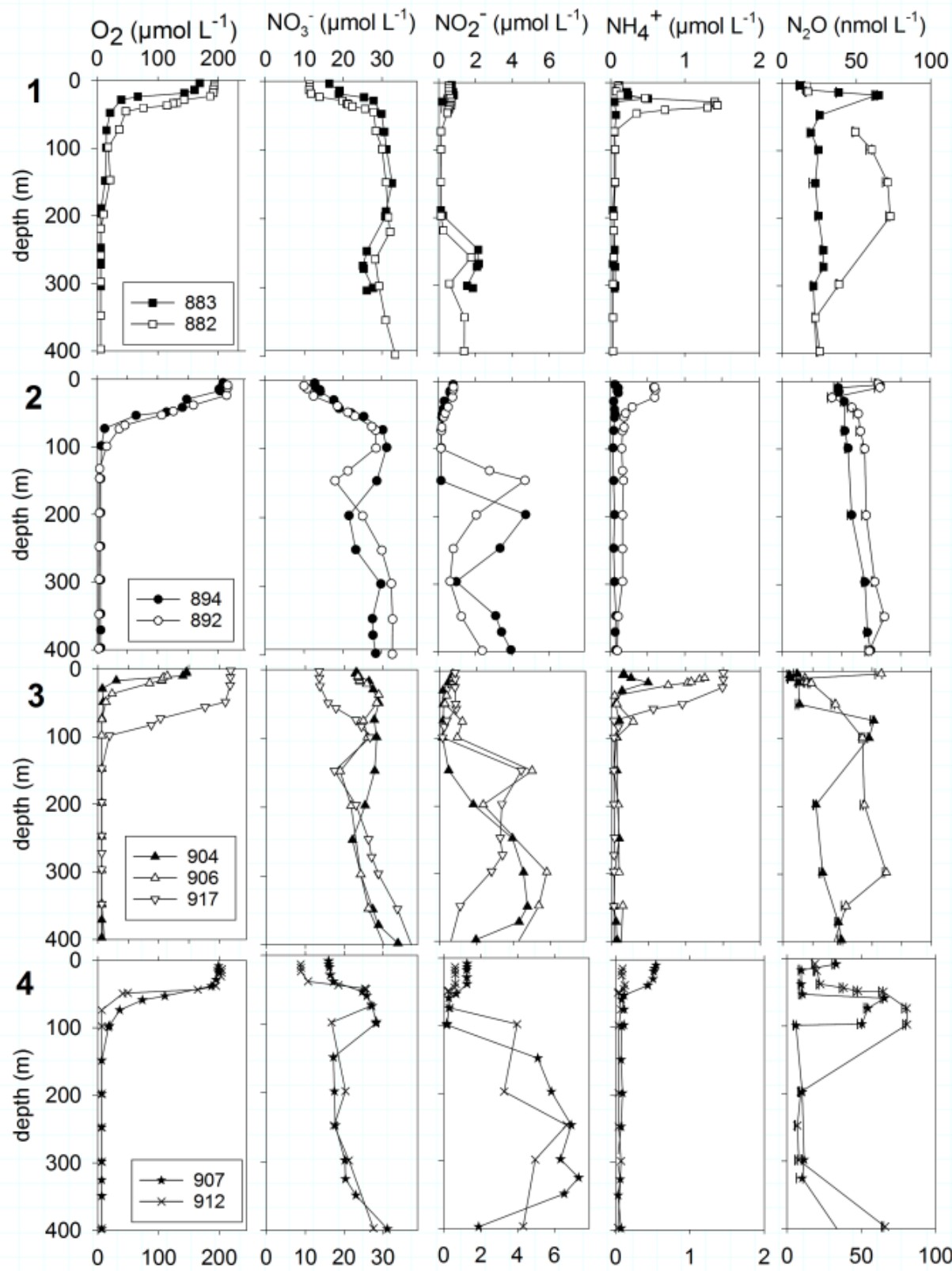

**Figure 2**

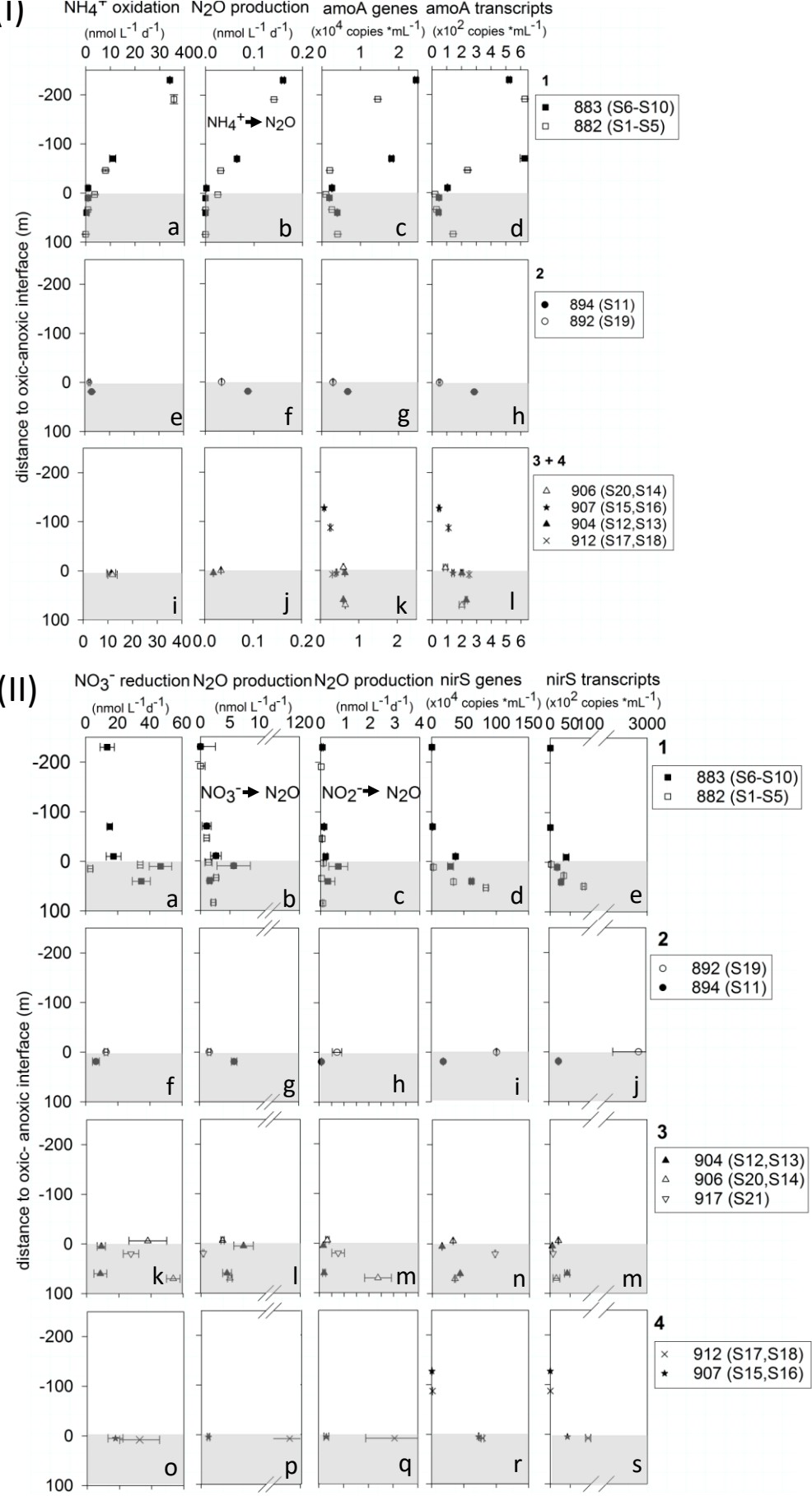

**Figure 3**

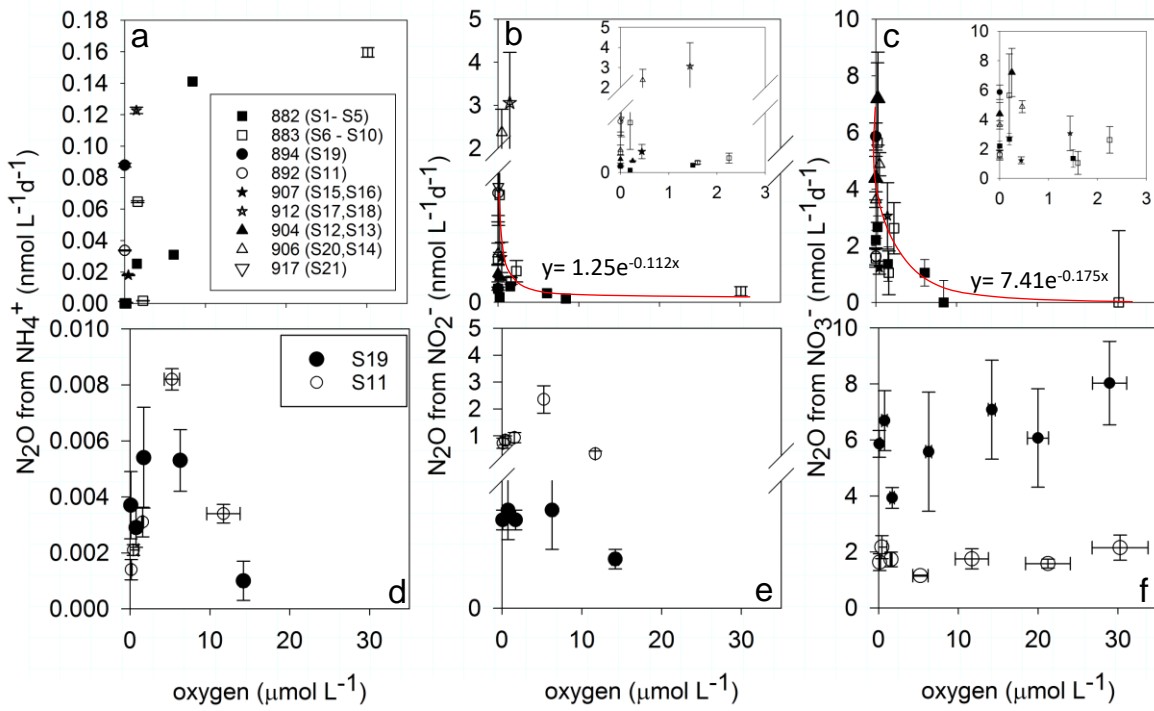

**Figure 4**

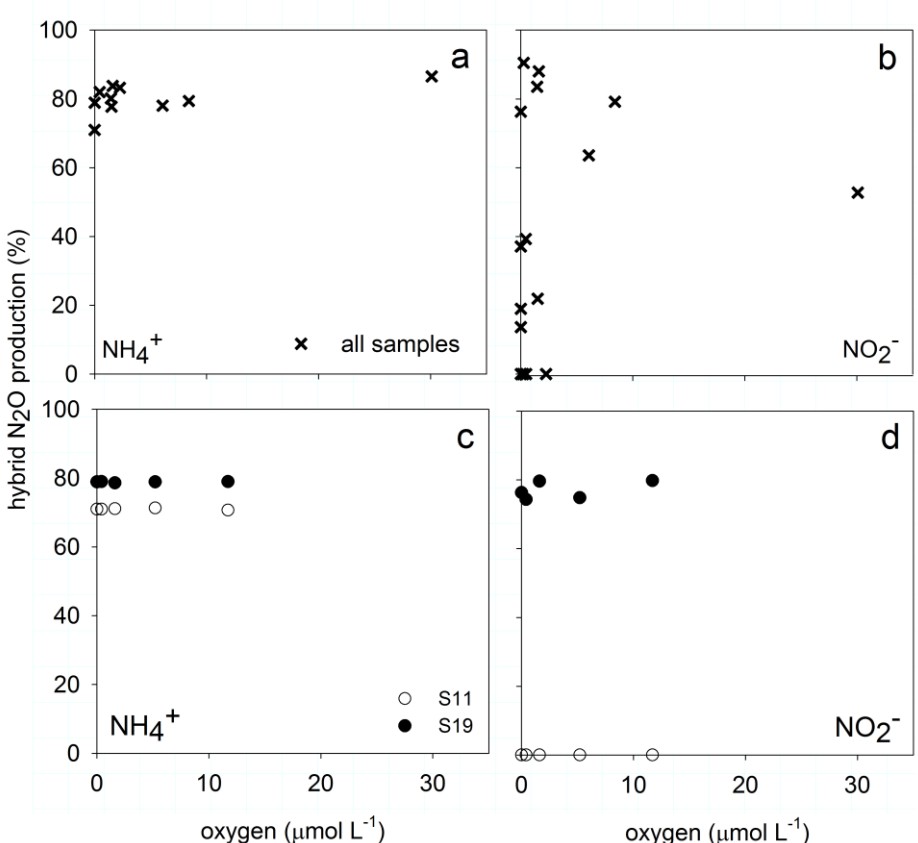

**Figure 5**

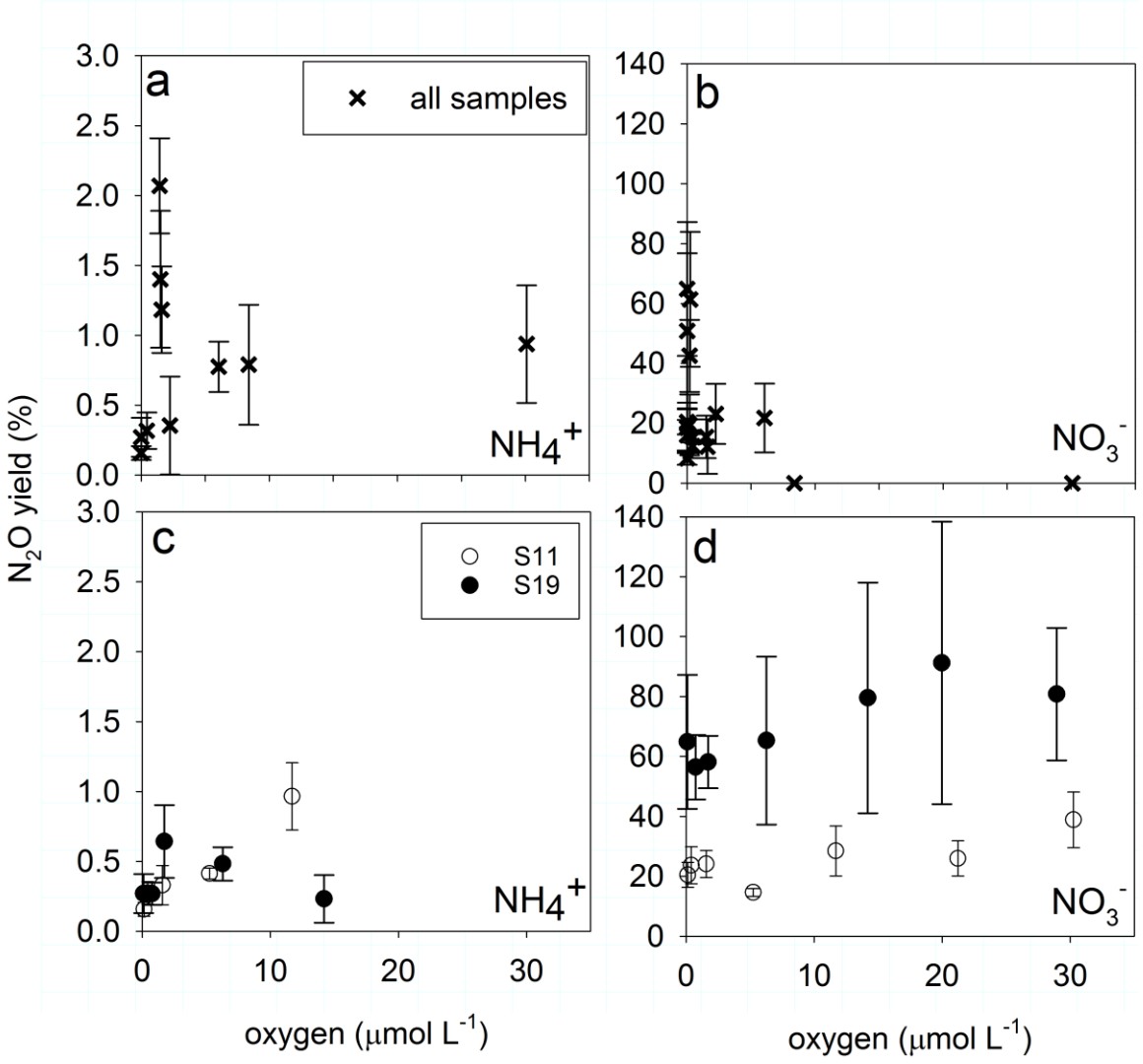

**Figure 6**

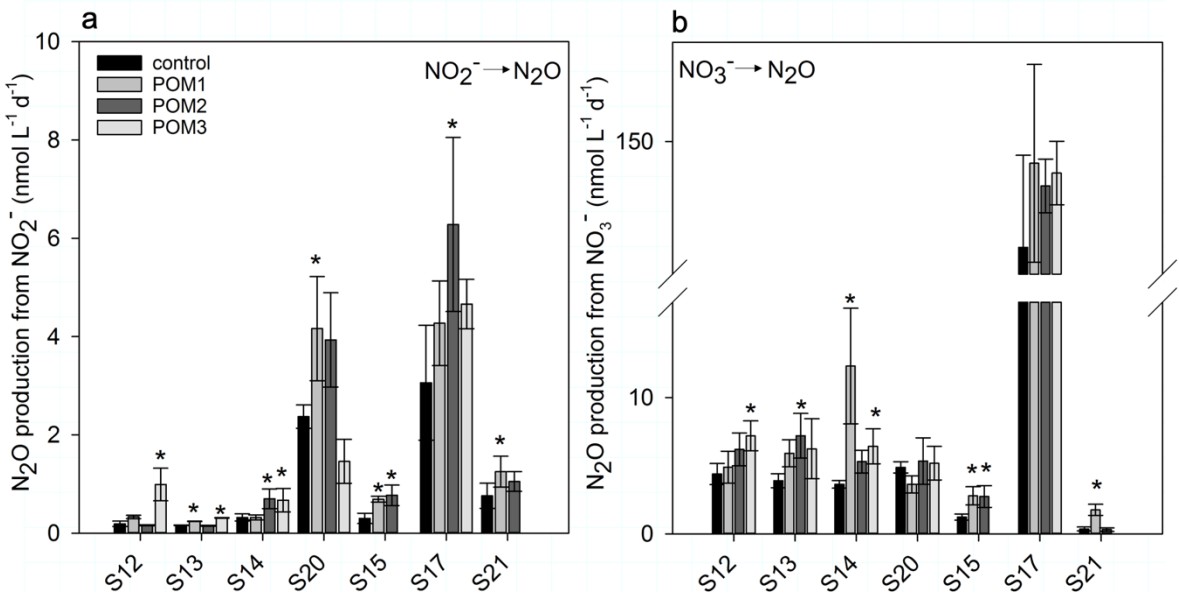

**Figure 7**

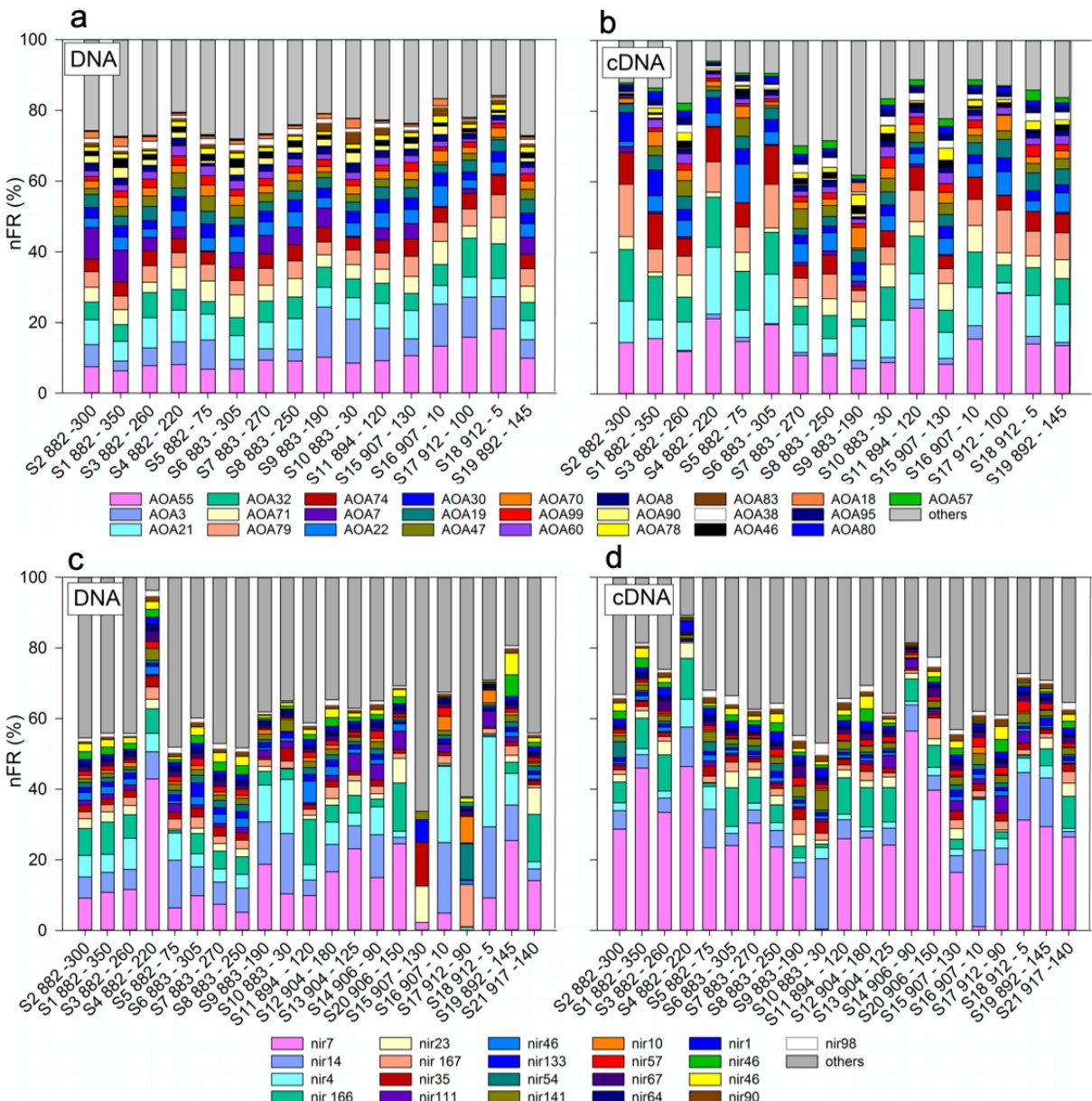

**Figure 8**

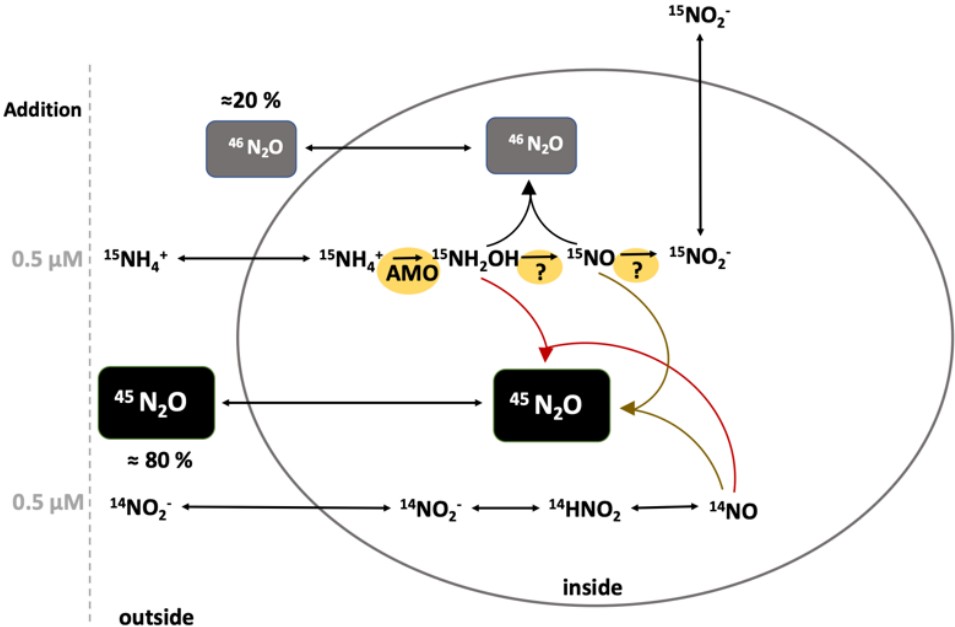

**Figure 9**