# Peer review of "Regulation of nitrous oxide production in low oxygen waters off the coast of Peru"

_Biogeosciences, 2019_

## Referee Comment (RC1) · Annie Bourbonnais (Referee) · 5 Jan 2020

**General comments:**
Frey et al. used a combined biogeochemical and microbial ecology approach to investigate $N_2O$ production in the ETSP off Peru. More specifically, they used 15N-labeled incubations to measure $N_2O$ production during ammonium oxidation as well as denitrification and their regulation by $O_2$ and particulate organic matter. They also measured *archaeal amoA* and *nirS* genes abundances and activity. Overall the manuscript is well written and will make a great contribution to the field as the exact mechanisms regulating $N_2O$ production in ODZs are still not well understood. Of particular importance, few studies have looked at the effects of organic matter addition on $N_2O$ production. I recommend publication after addressing the generally minor considerations below.

**Specific comments**
**Abstract:**

Another important finding is that hybrid $N_2O$ formation represented 70-86% of the $N_2O$ production during ammonium oxidation, regardless of the ammonium oxidation rate or $O_2$ concentrations. One sentence about this should be added to the abstract.

**Introduction**

Lines 70-75: The distinction between hybrid $N_2O$ production by ammonia oxidizing archaea and chemodenitrification (e.g. nitrite reduction coupled to iron II oxidation) should be better made. Hybrid $N_2O$ formation (mediated by AOA) has been observed in the ODZ water-column, but not chemodenitrification (also referred to as abiotic $N_2O$ production; Wankel et al., 2017), likely due to substrate limitation (Fe, Mn).

line 78: Correct nitrifier-denitrifiaction for denitrification.

Lines 79-81: It should be noted that Frame and Casciotti (2010) only observed higher yields at decreasing $O_2$ concentrations for high starting cell densities. At lower cell densities (closer to values found in ODZs), the impact of decreasing $O_2$ on $N_2O$ yield was much lower than observed in other studies.

Lines 102-104: Charpentier et al (2007) also suggested that nitrifier-denitrification is enhanced by high concentration of organic particles, which creates high $NO_2^-$ and low-$O_2$ microenvironments.

Lines 113-114: It would also be relevant to look at *nor* genes which are encoding nitric oxide reductase.

**Materials and methods:**

Line 136: It is not clear why a 3 mL He helium headspace is created before incubating, since it will impact in-situ $O_2$ concentrations.

Line136-137: I assume purging is done to avoid $O_2$ contamination? What is the $O_2$ threshold defining anoxia here? One potential problem with purging is that it also removes other gases (e.g., $H_2S$) involved in autotrophic denitrification (for instance, see Callbeck *et al*., 2018).

Lines 150-153: How did $O_2$ vary during the incubations? These data should perhaps be included as part of the supplementary materials.

Line 153: Explain the rationale for using particles >50 μm.

Lines 192-219: Plots showing increase in $^{15}N$ labeled products over time should be included in the supplementary materials. Were the relationships always linear?

Lines 228-229: These nirS primers exclude epsilon-proteobacteria (Murdock, et al., 2017). Epsilon proteobacteria are often the dominant portion of autotrophic sulfur oxidizers in sulfidic waters (e.g., Grote et al., 2008), thus this aspect should be discussed.

Line 256: Add accession number.

**Results:**

Lines 282-283: Could a contour plot of chlorophyll concentration added to the supplementary material for reference?

Lines 334-335: This result is a bit puzzling as previous studies (e.g., Dalsgaard et al., 2014), observed fifty percent inhibition of $N_2O$ production by denitrification at about 300 nM $O_2$. These observations are also unlike results from their *in situ* $O_2$ gradient experiments.

Lines 349-350: It is also surprising to observe the highest yield for $N_2O$ production at highest $O_2$ concentrations, for which $N_2O$ production should be inhibited (Dalsgaard et al., 2014).

**Discussion:**

Lines 421-426: Some of these are likely causal relationships.

Lines 425-426: This suggest that when $NO_3^-$ is abundant, denitrifying bacteria are less likely to use $NO_2^-$ (either from their internal pool or outside the cell) for $N_2O$ production during denitrification.

Line 441: What is the detection limit for $[N_2O]$?

Lines 441-444: Bourbonnais et al. (2017) used biogeochemical tracers ($N_2O$ concentrations and isotopes) that integrates over longer timescales compared to $^{15}N$-labeled incubations, which are more like taking a snapshot in time. Therefore, discrepancies between $N_2O$ production rate is expected and should be discussed in this context.

Line 451: Cite Fassbender et al. (2018) that discusses impacts of eddies on biogeochemical processes at different scales.

Lines 443: The error on this higher rate estimate seems rather large (in Figure 3, p).

Lines 458-460: This part is confusing. The $O_2$ threshold for reductive $N_2O$ production should be higher than for $N_2O$ consumption, not the converse. In other words, nitric oxide reductase should be more $O_2$ tolerant than nitrous oxide reductase (Dalsgaard et al., 2014). Otherwise, $N_2O$ would not accumulate.

Lines 445-446: I do not understand this statement.

Lines 479-481: This hypothesis is also supported by a rather long turnover time for $NO_2^-$ as inferred from the $\delta^{18}O$ of $NO_2^-$, which is generally fully equilibrated with water in offshore waters (Bourbonnais et al., 2015). This is not the case in coastal waters, where $NO_2^-$ seems to be more dynamic (see and cite Hu et al., 2016).

Lines 495-496: How can these contrasting results be reconciled?

Lines 522-524: If hybrid $N_2O$ formation during AOA is purely (or even partly) abiotic, then measured rates would be overestimated as $HgCl_2$ would not stop $N_2O$ production at the end of the incubations. For how long were these samples stored before being measured? This point should be better discussed.

Lines 565-566: What was the chlorophyll concentration in the center of the eddy?

Lines 641-643: $N_2O$ emission to the atmosphere are possible only if the water is upwelled.

Lines 649-652: Temporal variability is particularly not well captured in observational studies.

**Figure legends:**

Rename Figure 7: $N_2O$ production after additions of...

Figures 2 and 3 are too small. Legend (station #) is almost impossible to read.

Figure 5: Samples impacted by denitrification should be more clearly indicated (by a circle or rectangle and in the Figure legend) in Figure 5b.

**Supplements:**

Figures S1: I recommend expanding the scale at lower $O_2$ concentrations since this is the focus of the paper.

Figure S5: Add linear regression and r-square for natural samples in the zoom up plot.

Figure S6: Since there are only a few data points for $[NO_2^-]/[NO_3^-]$ higher than 0.10, I don't think the outlier (light gray dot) can be removed. There is much more scatter in Figure 5 in Ji et al. (2018) for the same relationship.

**Other references:**

Callbeck, C. M., Lavik, G., Ferdelman, T. G., Fuchs, B., Gruber-Vodicka, H. R., Hach, P. F., ... & Schunck, H. (2018). Oxygen minimum zone cryptic sulfur cycling sustained by offshore transport of key sulfur oxidizing bacteria. *Nature communications*, *9*(1), 1729.

Charpentier, J., Farias, L., Yoshida, N., Boontanon, N., & Raimbault, P. (2007). Nitrous oxide distribution and its origin in the central and eastern South Pacific Subtropical Gyre. *Biogeosciences Discussions*, *4*(3), 1673-1702.'

Fassbender, A. J., Bourbonnais, A., Clayton, S., Gaube, P., Omand, M., & Franks, P. J. S. (2018). Interpreting mosaics of ocean biogeochemistry. *Eos*, *99*(10.1029).

Hu, H., Bourbonnais, A., Larkum, J., Bange, H. W., & Altabet, M. A. (2016). Nitrogen cycling in shallow low-oxygen coastal waters off Peru from nitrite and nitrate nitrogen and oxygen isotopes. *Biogeosciences*, *13*(5), 1453-1468.

Murdock, S. A., & Juniper, S. K. (2017). Capturing compositional variation in denitrifying communities: a multiple-primer approach that includes Epsilonproteobacteria. *Appl. Environ. Microbiol.*, *83*(6), e02753-16.

---

## Referee Comment (RC2) · Anonymous Referee #2 · 15 Jan 2020

**Review of Frey et al; 'Regulation of nitrous oxide production in low oxygen waters off the coast of Peru'**

The authors combine 15N labelling experiments with analyses of gene abundances in the ODZ of the Peruvian coast in order to assess the pathways contributing to the production of the potent greenhouse gas, N2O. This is an interesting and overall well-written paper that adds to the relatively small amount of studies investigating the effects of small-scale O2 variations on N cycling processes in ODZs. However I would like to highlight some minor corrections/clarifications and several points in the (biogeochemical) methods that I think need to be shortly addressed/discussed in the paper before publication. Given the points already highlighted by R1, I will try not to repeat their comments

**General comments:**

Check nitrite/$NO_2^-$ throughout manuscript

You note differences in process rates and between the communities exposed *in situ* to O2 gradients and in the O2 manipulation experiments (e.g. Line 463-4). I think at least some discussion is needed as to the potential effects of purging the samples with gas as described in refs below (e.g. Dalsgaard et al, deBrabandere et al, Holtappels et al, Stewart et al.)

Can you be sure that there is no DNRA occurring in your experiments – in particular given the Lam et al. 2009 'Revising the N cycle…' paper also off the Peruvian coast. The presence of DNRA would complicate your isotope pairing experiments with 15NO3- and 15NO2- by transferring 15N into the NH4+ pool and you would get 'hybrid'N2O' of 15nh4+ and 15no2- forming 46N2O and be wrongly assigned. DNRA would also potentially dilute your 15NH4+ pool with 14N from background NO3- and alter the assumed 99% labelling in these experiments. I realise the contribution of AO to N2O production is small relative to denit, but the artefacts of DNRA on the rates/data should be discussed as it could lead to some N2O from AO being 'hidden'.

**Specific comments:**

Section 2.1: As with other papers with many sites, sampling points and manipulation experiments a written methods text quickly becomes very complicated with different additions, concentrations, replicates, time points etc. I think as a result of the text being quite confusing some information has been missed/is unclear. Adding a table of experiments, stations, variables, sampling routine (e.g. time points), number of replicates, other factors (e.g. whether O2 was measured in vials) would be informative/helpful to readers who are interested in comparing/replicating experiments.

Also Section 2.1: Missing info on NO3- and NO2- analyses (e.g. shown in Fig 2).

Line 145 (O2 manipulation experiments): Why was such a 'coarse' O2 range used compared to previous studies which use O2 manipulations generally below 1-2μM (e.g. Dalsgaard et al 2014, Bristow et al 2016)?

Line 145 (O2 manipulation experiments): This is a bit confusing: '…headspace volume was adjusted depending on the amount of site water added…'. Do you mean that after the addition of different oxygenated water volumes you also wanted to end up with a 3mL headspace as in the 'natural gradient' O2 experiment? Please rephrase and explain more clearly.

Line 153 (OM experiments): So only total N2O was measured in the OM experiments? Or were 15N substrates also added. Unclear as it is written now.

Line 166: Do you mean 'Ascarite' instead of Ascarid?

Line 186-8: Rephrase to: 'If more single labelled N2O is produced than expected (…), a hyrid formation of one nitrogen atom from nh4+ and one from no2- (…) is assumed to be taking place se found in archaeal ammonia oxidizers'

Line 191: What about 45N2O formed from dilution from background 14NO3- and 14NO2- in samples? Then you will get 45N2O from 15NOx + 14NOx … You note earlier (ca Line 140) that there is likely substantial 14NO3- (at least in some samples/depths) which will be reduced to 14NO2- and dilute your 15NO2- pool. Perhaps there is something I have missed in the text but this doesn't make sense to assume all 45N2O in incubations with 15NO2, especially in anoxic/low O2 manipulations where NOx can be respired.

Section 3.3: Could the % inhibition of processes be plotted to help comparison to other relevant studies on O2 manipulation on AO/no2- ox/denit (e.g. Kalvelage et al 2011, Dalsgaard et al 2014, Bristow et al 2016). I think at least some short discussion is warranted in relation to O2 effects on processes in these previous papers.

Section 3.4: This is a little confusing, additional 14NH4+ was also added along with the POC to experiments? Or was the POC filtered/rinsed after autoclaving?

Line 466: In relation to the 'Unchanged N2O production with higher O2 levels in NO3- treatments…' sentence: Can anoxic niches be ruled out in these experiments? You do note the sampling being during low upwelling and chl period but the settling of small particulates during experiments may create anoxic/low O2 zones to sustain anaerobic processes.

Sentence line 470-472 Bristow et al 2016 should also be a ref here in relation to kinetics of multi-step processes

Line 477: How can you be sure none of the N2O was consumed without further measurements (e.g. 15N-N2)? Production may just be much faster than consumption.

Line 512-515: Confusing sentences, consider rephrasing.

Line 521: This is a bit of an oversimplification - because something is below detection doesn't necessarily mean nothing is happening, more likely a tight coupling between consumption and production (e.g. see Figure 4 in Klawonn et al 2019 and Figure 3 in Olofsson et al 2019 references). Could there be a dilution of your 15NH4+ pool to consider due to rapid cryptic cycling on shorter scales than your experiments? Ideally 15NH4+ and total NH4+ would be followed through the time series to check for dilution effects. Both show very rapid NH4+ turnover (within ~5h) in oligotrophic waters

Line 532: If measured, the accumulation and consumption of intermediates (e.g. NO2-) could also be used to imply biotic vs abiotic mechanisms (e.g. Betlach and Tiedje 1981 reference).

Line 560-3: Could a 15N recovery/inventory be calculated for the experiments (e.g. 15N recovery from initial substrate, measured intermediates and 15N-N2O?) This could help infer a % N2O production from denitrification which is important for putting the N2O production from denit in context – i.e. how do variations in O2 impact the proportion of N2O produced by denit relative to N2?

Fig 4 b, c & Fig 6 b: consider zoomed-in insert of x-axis (e.g. similar to Fig S5)

Figure S5: Seems to be two different slopes here from manipulated vs natural O2 gradients – could also be discussed in relation to purging artefacts.

**Ref suggestions**

Betlach, M. R., and J. M. Tiedje. 1981. Kinetic explanation for accumulation of nitrite, nitric oxide, and nitrous oxide during bacterial denitrification. Appl. Environ. Microbiol. 42: 1074–1084.

Bristow et al 2016 Ammonium and nitrite oxidation at nanomolar oxygen concentrations in oxygen minimum zone waters. PNAS

Dalsgaard, T. et al. Oxygen at nanomolar levels reversibly suppresses process rates and gene expression in anammox and denitrification in the oxygen minimum zone off Northern Chile. mBio 5,

De Brabandere, L., Thamdrup, B., Revsbech, N. P. & Foadi, R. A critical assessment of the occurrence and extend of oxygen contamination during anaerobic incubations utilizing commercially available vials. J. Microbiol. Methods 88, 147–154 (2012).

Holtappels, M., Lavik, G., Jensen, M. M. & Kuypers, M. M. M. in Methods in Enzymology: Research on Nitrification and Related Processes, Part A, Vol. 486 (ed. Klotz, M. G.) 223–251

Kalvelage et al 2011' Oxygen sensitivity of anammox and coupled N-cycle processes in oxygen minimum zones.' PlosOne

Klawonn et al 2019 'Untangling hidden nutrient dynamics: rapid ammonium cycling and single-cell ammonium assimilation in marine plankton communities' ISME

Olofsson et al 2019 Nitrate and ammonium fluxes to diatoms and dinoflagellates at a single cell level in mixed field communities in the sea. Scientific Reports

Stewart et al 2012 'Experimental Incubations Elicit Profound Changes in Community Transcription in OMZ Bacterioplankton' PlosOne

---

## Referee Comment (RC3) · Anonymous Referee #2 · 24 Jan 2020

One more reference suggestion I just came across (although I'm quite sure that the authors are already aware of the paper) with regards to artefacts of sampling from low O2 waters:

Torres-Beltrán et al 2019 'Sampling and Processing Methods Impact Microbial Community Structure and Potential Activity in a Seasonally Anoxic Fjord: Saanich Inlet, British Columbia' Frontiers in Marine Science

---

## Author Comment (AC1) · 31 Jan 2020

**Interactive comment "Regulation of nitrous oxide production in low oxygen waters off the coast of Peru"**

Response to Referee #1:

We are grateful to the reviewer for the positive feedback and constructive suggestions which greatly helped us in preparing a revised manuscript.
We addressed the specific suggestions below (our replies in bold).

Abstract: Another important finding is that hybrid $N_2O$ formation represented 70-86% of the $N_2O$ production during ammonium oxidation, regardless of the ammonium oxidation rate or $O_2$ concentrations. One sentence about this should be added to the abstract.

**We added: "Hybrid $N_2O$ formation (i.e. $N_2O$ getting one N atom from $NH_4^+$ and the other from other substrates such as $NO_2^-$) was the dominant species, comprising 70 – 86 % of total produced $N_2O$ from $NH_4^+$, regardless of the ammonium oxidation rate or $O_2$ concentrations."**

Introduction Lines 70-75: The distinction between hybrid $N_2O$ production by ammonia oxidizing archaea and chemodenitrification (e.g. nitrite reduction coupled to iron II oxidation) should be better made. Hybrid $N_2O$ formation (mediated by AOA) has been observed in the ODZ water-column, but not chemodenitrification (also referred to as abiotic $N_2O$ production; Wankel et al., 2017), likely due to substrate limitation (Fe, Mn).

**In line 74, we added: Abiotic $N_2O$ production, also known as chemodenitrification, from intermediates like $NH_2OH$, NO or $NO_2^-$ can occur under acidic conditions (Frame et al. 2017), or in the presence of reduced metals like Fe or Mn and catalyzing surfaces (Zhu-Barker et al. 2015, Wankel et al. 2017), but the evidence of abiotic $N_2O$ production (chemodenitrification) in ODZs is still lacking.**

line 78: Correct nitrifier-denitrifiaction for denitrification.

**We added (line 80) denitrification, but did not replace nitrifier-denitrification because this is what the paragraph is about.**

Lines 79-81: It should be noted that Frame and Casciotti (2010) only observed higher yields at decreasing $O_2$ concentrations for high starting cell densities. At lower cell densities (closer to values found in ODZs), the impact of decreasing $O_2$ on $N_2O$ yield was much lower than observed in other studies.

**We re-wrote the sentence as follow (line 83-87): Overall, the yield of $N_2O$ per $NO_2^-$ generated from AO is lower in AOA then AOB (Hink et al. 2017, 2018) but it should be noted that the degree to which $N_2O$ yield increases with decreasing $O_2$ concentrations is variable with cell densities in cultures or field sites, (Cohen & Gordon 1978; Yoshida 1988; Goreau et al. 1980; Frame & Casciotti 2010, Santoro et al. 2011, Löscher et al. 2012, Ji et al. 2015a, 2018a).**

Lines 102-104: Charpentier et al (2007) also suggested that nitrifier-denitrification is enhanced by high concentration of organic particles, which creates high $NO_2^-$ and low-$O_2$ microenvironments.

**We added a sentence in line 81-83. "It has also been suggested that high concentration of organic particles create high $NO_2^-$ and low-$O_2$ microenvironments enhancing nitrifier-denitrification (Charpentier et al. 2007)."**

Lines 113-114: It would also be relevant to look at *nor* genes which are encoding nitric oxide reductase.

**The reviewer is correct, but the goal here was to distinguish between nitrifiers and denitrifiers and for that the *nor* gene is not ideal as it is present in both. Furthermore, in Fuchsmann et al. (2017) (doi10.3389/fmicb.2017.02384) the canonical forms of the gene *norB* and *qnorB* were very low abundant, suggesting that there might be other genes encoding enzymes mediating NO reduction to $N_2O$.**

Materials and methods:
Line 136: It is not clear why a 3 mL He helium headspace is created before incubating, since it will impact in-situ $O_2$ concentrations.

**The headspace was added for several reasons: 1) to avoid diffusion of oxygen from the septum into the liquid directly, headspace provided an additional barrier, 2) to be able to purge the serum bottles and 3) to avoid artificial differences by different treatments, all bottles received a headspace. We added (line 148 – 150): "He purging removed dissolved oxygen contamination which is likely introduced during sampling and the headspace prevents direct oxygen leakage from the rubber seals (DeBrabandere et al. 2012)."**

Line136-137: I assume purging is done to avoid $O_2$ contamination? What is the $O_2$ threshold defining anoxia here? One potential problem with purging is that it also removes other gases (e.g., $H_2S$) involved in autotrophic denitrification (for instance, see Callbeck *et al.*, 2018).

**Yes, purging was done to decrease oxygen contamination during sampling. We rewrote the sentence as such (line 147-148): A 3 mL helium (He) headspace was created and samples from anoxic ($O_2$ < below detection) water depths were He purged for 15min. We also added line 148 – 150, as written as answer to your previous comment. The point of H2S removal during purging is added into the discussion section line 511- 521: "In addition, sampling with Niskin bottles and purging can induce stress responses (Stewart et al. 2012) and shift the richness and structure of the microbial community from the *in situ* community (Torres-Beltran et al. 2019), which can be one potential explanation for the different responses between manipulated $O_2$ and *in situ* $O_2$ experiments. The removal of other gases like $H_2S$ during purging introduces another potential artefact. However, this is unlikely because measurable $H_2S$ concentrations have mostly been found at very shallow coastal stations (< 100 m deep) (Callbeck et al. 2018), which have not been sampled in this study. On the contrary, high abundances (up to 12 %) of sulfur oxidizing gamma proteobacteria, like SUP05 can be found in eddy-transported offshore waters where they**

**actively contributed to autotrophic denitrification (Callbeck et al. 2018). This study cannot differentiate between autotrophic or organotrophic denitrification, but a contribution of autotrophic denitrification in the eddy center is likely."**

Lines 150-153: How did $O_2$ vary during the incubations? These data should perhaps be included as part of the supplementary materials.

**The oxygen concentrations stayed constant in the low oxygen treatments, while it decreased in higher oxic treatments. That explains the higher standard deviation in higher treatments. Oxygen concentrations over time are added to the supplements Figure S1.**

Line 153: Explain the rationale for using particles >50 µm.

**It is the fraction that is sinking. This is stated in line 607-608: "However, the particle size (>50 µm) used in the experiments is indictive of sinking particles."**

Lines 192-219: Plots showing increase in $_{15}N$ labeled products over time should be included in the supplementary materials. Were the relationships always linear?

**Linear relationships were used to calculate the slopes and only significant slopes were included as written in line 233-235. We added example time plots from the oxygen manipulation experiments into the supplements. See Figure S2.**

Lines 228-229: These nirS primers exclude epsilon-proteobacteria (Murdock, et al., 2017). Epsilon proteobacteria are often the dominant portion of autotrophic sulfur oxidizers in sulfidic waters (e.g., Grote et al., 2008), thus this aspect should be discussed.

**We added a statement in the methods that we are aware that epsilon-proteobacteria are not captured with the Primer we used. Line 260 - 263: "The *nirS* Primers are not specific for epsilon-proteobacteria (Murdock et al. 2017), but in previous metagenomes from the ETSP epsilon-proteobacteria where below 3-4% or not found, except in very sulfidic, coastal stations (Stewart et al. 2011, Wright et al. 2012, Ganesh et al. 2012, Schunck et al. 2013, Kavelage et al. 2015)."**

Line 256: Add accession number.

**Added. GEO Accession No GSE142806**

**Results:**

Lines 282-283: Could a contour plot of chlorophyll concentration added to the supplementary material for reference?

**We added surface Chlorophyll data to the station map. See Figure 1.**

Lines 334-335: This result is a bit puzzling as previous studies (e.g., Dalsgaard et al., 2014), observed fifty percent inhibition of N2O production by denitrification at about 300nM O2. These observations are also unlike results from their *in situ* O2 gradient experiment.

**This is not contradictory to Dalsgaard et al. 2014. They were in depths with high NO2- concentration indicative for the core of the anoxic zone, whereas this study took place at the upper part of the anoxic zone and in the oxycline.**

Lines 349-350: It is also surprising to observe the highest yield for $N_2O$ production at highest $O_2$ concentrations, for which $N_2O$ production should be inhibited (Dalsgaard et al., 2014).

**This is not to be confused with the N2O yield/N2. The yields are for NO3- and not like in Dalsgaard with 15NO2- which apparently makes a large difference as we can show in this study!!!**

**Discussion:**

Lines 421-426: Some of these are likely causal relationships.

**Yes, absolutely.**

Lines 425-426: This suggest that when $NO_3-$ is abundant, denitrifying bacteria are less likely to use $NO_2-$ (either from their internal pool or outside the cell) for $N_2O$ production during denitrification.

**This comment is added to the text line 460 - 461.**

Line 441: What is the detection limit for $[N_2O]$?

**The detection limits is 2nM. We added that information into the method section 2.1. line 137.**

Lines 441-444: Bourbonnais et al. (2017) used biogeochemical tracers ($N_2O$ concentrations and isotopes) that integrates over longer timescales compared to $_{15}N$-labeled incubations, which are more like taking a snapshot in time. Therefore, discrepancies between $N_2O$ production rate is expected and should be discussed in this context.

**We rewrote that section: "Previously reported maximum rates were up to 86 nmol $L^{-1}$ $d^{-1}$ (Dalsgaard et al. 2012) based on $^{15}N$ tracer incubations. Much smaller maximum rates, 49 nmol $L^{-1}$ $d^{-1}$ (Bourbonnais et al. 2017) and 50 nmol $L^{-1}$ $d^{-1}$ (Farìas et al. 2009), were obtained using $N_2O$ isotope and isotopomer approaches which provide time and process integrated signals. Hence, the deviation of maximum rates can be explained by 1) the different approaches and 2) the sampling of the core of the eddy. "**

Line 451: Cite Fassbender et al. (2018) that discusses impacts of eddies on biogeochemical processes at different scales.

**We did not add Fassbender here, because the recommended paper does not contain information on impact of eddy age on the N2O distribution, which is the point we are trying to make here.**

Lines 443: The error on this higher rate estimate seems rather large (in Figure 3, p).

**We added the exact rate with the standard deviation.**

Lines 458-460: This part is confusing. The $O_2$ threshold for reductive $N_2O$ production should be higher than for $N_2O$ consumption, not the converse. In other words, nitric oxide reductase should be more $O_2$ tolerant than nitrous oxide reductase (Dalsgaard et al., 2014). Otherwise, $N_2O$ would not accumulate.

**This is exactly my point. There is a discrepancy between the thresholds in rates we find and the N2O concentration maxima we measure between 1 – 8uM O2. If N2O production is so sensitive from denitrification then where is all the N2O coming from? Just NH4+ oxidation is unlikely based on the N2O production rates we find from NH4+ oxidation. There might be a higher threshold for N2O production from denitrification?**

Lines 445-446: I do not understand this statement.

**We did not measure N2 production rates, so we cannot say anything about the N2O/N2 yield during denitrification. This yield is subject to changes and not constant, Because of that, we have no chance to make an estimate on the N2 production rate. Maybe in the Eddy incomplete denitrification to N2O was favored and that is what we measured or complete denitrification was fueled and this is what we measured. We rephrased the sentence (line 481- 485) to "N₂ production measurements were not performed in this study, so it cannot be determined whether the eddy only stimulated N₂O production but not N₂ production from denitrification (i.e. increasing the N₂O/N₂ yield) or if the eddy also increased complete denitrification to N₂ by 10 times compared to stations outside of the eddy. "**

Lines 479-481: This hypothesis is also supported by a rather long turnover time for $NO_2^-$ as inferred from the $\delta_{18}O$ of $NO_2^-$, which is generally fully equilibrated with water in offshore waters (Bourbonnais et al., 2015). This is not the case in coastal waters, where $NO_2^-$ seems to be more dynamic (see and cite Hu et al., 2016).

**We added this statement into the manuscript as follow (line 532 – 534): "Long turnover times for NO₂⁻ have been inferred from d¹⁸O of NO₂⁻, which was fully equilibrated with water in the offshore waters (Bourbonnais et al. 2015) and more dynamic in the coastal waters (Hu et al. 2016) supporting our hypothesis. "**

Lines 495-496: How can these contrasting results be reconciled?

**We attribute this to the intensity of the ammonium oxidation rate which exerts a first order control on the N2O production rate. Meaning if the NH4+ oxidation rates would exponential decrease with O2 concentration then we would find that relationship in the N2O production rates. We discuss this further down in line 554 – 556.**

Lines 522-524: If hybrid N2O formation during AOA is purely (or even partly) abiotic, then measured rates would be overestimated as HgCl2 would not stop N2O production at the end of the incubations. For how long were these samples stored before being measured? This point should be better discussed.

**The samples were stored between 2 – 5 month. Abiotic N2O production would take place and continue until we measure the samples, indeed. But it also goes on in all samples raising the N2O baseline (in mass 44,45,46) for all and not just in specific ones. This impact will likely vary with depth, but then all the timepoints are affected by the same abiotic production. The rates are calculated from the increase over time making them independent of the baseline. We added a figure to the supplements S9, where results for abiotic production from 15NO2- tracer are shown from 4 depths from 2 stations. The addition of 15NO2- tracer results in little abiotic production; 0.018 – 0.37 nM 45N2O and 0.009 – 0.026 nM 46N2O up to the point of mass spec analysis, but independent HgCl2 addition. We added this point into the discussion line 281. However, we did not test abiotic N2O from NH4+ tracer, hence this can not be fully ruled out. We added that point in line 588-590: "Additionally, at four depths the potential for abiotic $N_2O$ production in $^{15}NO_2^-$ addition experiments showed variations with depth and no significant impact of HgCl2 fixation (Figure S9)."**

Lines 565-566: What was the chlorophyll concentration in the center of the eddy?

**Low, below 1mg/m3. We added a map with surface Chlorophyll, see Figure 1.**

Lines 641-643: N2O emission to the atmosphere are possible only if the water is upwelled.

**We rephrased the sentence to (line 698): "Regardless of which processes are responsible for $N_2O$ production in the ODZ, high $N_2O$ production at the oxic-anoxic interface of the upper oxycline sustains high $N_2O$ concentration peaks with a potential for intense $N_2O$ emission to the atmosphere during upwelling events."**

Lines 649-652: Temporal variability is particularly not well captured in observational studies.

**We added a sentence to pick up on that comment (line 705 – 706): "While this study does not help to resolve temporal variability, manipulation experiments give valuable insights on the short-term response of $N_2O$ production to oxygen and particles. "**

Figure legends:
Rename Figure 7: N2O production after additions of...
      **The figure was renamed accordingly.**

Figures 2 and 3 are too small. Legend (station #) is almost impossible to read.

      **The figure Legend and axis label were adjusted.**

Figure 5: Samples impacted by denitrification should be more clearly indicated (by a circle or rectangle and in the Figure legend) in Figure 5b.

**In all samples in Figure 5b, N$_2$O production from 15NO3- was found. If that is what the reviewer means. There was no adjustment done to the figure.**

Supplements:
Figures S1: I recommend expanding the scale at lower O$_2$ concentrations since this is the focus of the paper.

**We did not expand the scale here as the focus is the shallowing of the oxycline in the center of the eddy , which is nicely visible in this figure.**

Figure S5: Add linear regression and r-square for natural samples in the zoom up plot.

**Linear regressions and equations were added to the Figure S7.**

Figure S6: Since there are only a few data points for [NO$_2$-]/[NO$_3$-] higher than 0.10, I don't think the outlier (light gray dot) can be removed. There is much more scatter in Figure 5 in Ji et al. (2018) for the same relationship.

**The point was included into the regression.**

---

## Author Comment (AC2) · 31 Jan 2020

Response to Referee #2:
We thank to reviewer for the positive feedback and valuable suggestions which greatly improved the quality of the manuscript.
We addressed the specific suggestions below (our replies in bold).

*General comments:*

Check nitrite/NO $^-$ throughout manuscript 2

**We checked for consistency and changed all nitrites to NO$_2$$^-$.**

You note differences in process rates and between the communities exposed *in situ* to O2 gradients and in the O2 manipulation experiments (e.g. Line 463-4). I think at least some discussion is needed as to the potential effects of purging the samples with gas as described in refs below (e.g. Dalsgaard et al, deBrabandere et al, Holtappels et al, Stewart et al.)

**We added a sentence to the method section, line 148-151: "He purging removed dissolved oxygen contamination which is likely introduced during sampling and the headspace prevents possible oxygen leakage from the rubber seals (DeBrabandere et al. 2012)" and in line 158/159: "Total incubation times were adjusted to prevent bottle effects, which become significant after 20 h based on respiration rate measurements (Tiano et al. 2014). "**
**Furthermore, we added a part in the discussion line 508-521. "Different responses of N$_2$O production rates to O$_2$ between *in situ* assemblages and incubation were not unexpected because different rates at different depths were likely not only due to O$_2$ differences but also other factors such as different organic matter fluxes and different amounts and types of N$_2$O producers at different depths. In addition, sampling with Niskin bottles and purging can induce stress responses (Stewart et al. 2012) and shift the richness and structure of the microbial community from the *in situ* community (Torres-Beltran et al. 2019), which can be one potential explanation for the different responses between manipulated oxygen and *in situ* oxygen experiments. The removal of other gases like H$_2$S during purging introduces another potential artefact. However, it is unlikely as measurable H$_2$S concentrations have mostly been found at very shallow coastal stations (< 100 m deep) (Callbeck et al. 2018), which was not the case in this study. On the contrary, high abundances (up to 12%) of sulfur oxidizing gamma proteobacteria, like SUP05 can be found in eddy-transported offshore waters where they actively contributed to autotrophic denitrification (Calbeck et al. 2018). In this study, it cannot differentiate between autotrophic or organotrophic denitrification, but a contribution of autotrophic denitrification in the eddy center is likely."**

Can you be sure that there is no DNRA occurring in your experiments – in particular given the Lam et al. 2009 'Revising the N cycle...' paper also off the Peruvian coast. The presence of DNRA would complicate your isotope pairing experiments with 15NO3- and 15NO2- by transferring 15N into the NH4+ pool and you would get 'hybrid'N2O' of 15nh4+ and 15no2- forming 46N2O and be wrongly assigned. DNRA would also potentially dilute your 15NH4+ pool with 14N from background NO3- and alter the assumed 99% labelling in these experiments. I realise the contribution of AO to N2O production is small relative to denit, but the artefacts of DNRA on the rates/data should be discussed as it could lead to some N2O from AO being 'hidden'.

**The reviewer raises a very important point and no, we cannot be sure that the occurrence of DNRA is impacting our results. We added this consideration to the manuscript in line 216 - 232. "Nevertheless, this assumption brings some initial considerations which need to be accounted for. There is a potential for overestimating hybrid $N_2O$ production in $^{15}NO_2^-$ incubations by 5% in samples with high $NO_3^-$ reduction rates. But in incubations from anoxic depths with high $NO_3^-$ reduction rates, no hybrid $N_2O$ production is found at all. For example, accounting for a decrease in $f_N$ of the $NO_3^-$ pool by active $NO_2^-$ oxidation, the process with highest rates (Sun et al. 2017), had an effect of only $\pm 0.2$ % on the final rate estimate. The presence of DNRA complicates $^{15}N$-labelling incubations because it can change f in all three tracer experiments. In the vicinity of DNRA in $^{15}NO_3^-$ incubations, $^{15}NO_2^-$ and $^{15}NH_4^+$ can be produced from $^{15}NO_3^-$ which can contribute to $^{46}N_2O$ production by AO. Even when a maximum DNRA rate (20 nM $d^{-1}$ in Lam et al. 2009) is assumed to produce 0.02 nM $^{15}NH_4^+$ in 24 h with all of it being oxidized to $N_2O$ (max. $N_2O$ production from AO 0.16 nM $d^{-1}$, this study), its contribution to $^{46}N_2O$ production is likely minor and within the standard error of $N_2O$ production rates from $NO_3^-$. Hence an overestimation of the $N_2O$ production rates is unlikely. The same applies in incubations with $^{15}N$-$NO_2^-$ when DNRA produces $^{15}NH_4^+$, additional $^{46}N_2O$ can be produced with a hybrid mechanism by AO not accounted for in the present rate calculations. In $^{15}NO_2^-$ incubations with high starting f (>0.7) the production of $^{14}NO_2^-$ by $NO_3^-$ reduction (which decreases f) leads to an underestimation by max. 9%, whereas in incubations with a low f (<0.3) the effect is less with max. 3 % underestimation of $N_2O$ production rates. In $^{15}NH_4^+$ incubations (f >0.9), max. DNRA rate would lead to an underestimation of 3.5 %. "**

**Specific comments:**

Section 2.1: As with other papers with many sites, sampling points and manipulation experiments a written methods text quickly becomes very complicated with different additions, concentrations, replicates, time points etc. I think as a result of the text being quite confusing some information has been missed/is unclear. Adding a table of experiments, stations, variables, sampling routine (e.g. time points), number of replicates, other factors (e.g. whether O2 was measured in vials) would be informative/helpful to readers who are interested in comparing/replicating experiments.

**It is correct, that such set ups can get confusing very quickly, but in table 1 stations, depths, measured variables and the kind of experiment performed are given. However, we added one column with the kind of tracer addition we did. The replicates and time points did not vary between experiments and hence is only stated in the test. We only measured oxygen in one bottle with each incubation per depth or treatment, which was also consistent and written in the text line 168/169.**

Also Section 2.1: Missing info on NO3- and NO2- analyses (e.g. shown in Fig 2).

**The measurement of nitrite and nitrate concentrations is given in line 130-132.**

Line 145 (O2 manipulation experiments): Why was such a 'coarse' O2 range used compared to previous studies which use O2 manipulations generally below 1-2μM (e.g. Dalsgaard et al 2014, Bristow et al 2016)?

**In Bristow et al. 2016 a and b the maximal oxygen concentration in their manipulation experiments was 10uM and 20uM dissolved oxygen, so we are not quite sure what the referee means. Dalsgaard et al 2014 performed a really nice microcosm experiment, where oxygen concentrations were monitored online in the flask they subsampled. In our case, each time point was a separate bottle making it impossible to use such an approach. For the experimental design in this study, it was important to choose oxygen levels where we can be sure that oxygen concentrations are different enough from each other that we can differentiate the two treatments (f.e. 100nM and 200nM would be tricky to tell apart with our standard deviations of 180nM and 240nM over 24h). We added a plot of oxygen over time into the supplements Figure S1.**

Line 145 (O2 manipulation experiments): This is a bit confusing: '...headspace volume was adjusted depending on the amount of site water added...'. Do you mean that after the addition of different oxygenated water volumes you also wanted to end up with a 3mL headspace as in the 'natural gradient' O2 experiment? Please rephrase and explain more clearly.

**Yes, that is exactly what we were trying to do. The sentence was rephrased (line 163 – 164) to "For the $O_2$ manipulation experiments, all serum bottles were He purged and after the addition of different amounts of air saturated site water a final headspace volume of 3 mL was achieved."**

Line 153 (OM experiments): So only total N2O was measured in the OM experiments? Or were 15N substrates also added. Unclear as it is written now.

**$^{15}$N substrates were also added in the organic matter addition experiments. We changed the text to : "For all experiments,.." in line 151 and adjusted line 174 as followed: "200µL of POC solution were added to each serum bottle before $^{15}$N-NO$_3^-$ or $^{14}$N-NO$_2^-$ tracer injection."**

Line 166: Do you mean 'Ascarite' instead of Ascarid?

**Yes, we mean "Ascarite" and it was changed.**

Line 186-8: Rephrase to: 'If more single labelled N2O is produced than expected (...), a hyrid formation of one nitrogen atom from nh4+ and one from no2- (...) is assumed to be taking place se found in archaeal ammonia oxidizers'

**The sentence was rephrased as recommended.**

Line 191: What about 45N2O formed from dilution from background 14NO3- and 14NO2- in samples? Then you will get 45N2O from 15NOx + 14NOx ... You note earlier (ca Line 140) that there is likely substantial 14NO3- (at least in some samples/depths) which will be reduced to 14NO2- and dilute your 15NO2- pool. Perhaps there is something I have missed in the text but this doesn't make sense to assume all 45N2O in incubations with 15NO2, especially in anoxic/low O2 manipulations where NOx can be respired.

**Indeed, there is a potential for overestimating hybrid N2O production in 15NO2-incubations by 5% in samples with high NO3- reduction rates. But in incubations from anoxic depths with high NO3- reduction rates, no hybrid N2O production is found at**

**all. We added all potential problems which come with the assumption of a costant f into that section, starting line 217.**

Section 3.3: Could the % inhibition of processes be plotted to help comparison to other relevant studies on O2 manipulation on AO/no2- ox/denit (e.g. Kalvelage et al 2011, Dalsgaard et al 2014, Bristow et al 2016). I think at least some short discussion is warranted in relation to O2 effects on processes in these previous papers.

[Figure]

**We added the inhibition curves along the natural O2 gradient here, but we stay with the same figures in the main text as we want to show absolute rates. Section 3.3 is a results section, so we do not refer to papers there. However, in the discussion part 4.1 we cited Kalvelage et al. 2011 (line 525) and Dalsgaard et al. 2014 (line 522). Bristow et al. 2016 is added to the section 4.1. Out of these papers, only Dalsgaard et al. 2014 measured N2O production, hence the focus in the discussion is on their paper.**

Section 3.4: This is a little confusing, additional 14NH4+ was also added along with the POC to experiments? Or was the POC filtered/rinsed after autoclaving?

**The particles were in 0.2um filtered low nutrient seawater and during autoclaving some nitrogen from the particles was liberated into that solution. By adding 200uL of that concentrated POM solution with max. 1.56µM of NH4+ means we added minor amounts of NH4+ into our incubation bottles (0.3nM NH4+), which is neglectable form that perspective. It may be more important with respect to the change in N/C ratios. We adjusted the text, which reads now (line 386 – 389): "The autoclaving of the concentrated POM solution liberated $NH_4^+$ from the particles, reducing the N/C ratio of the particles compared to non-autoclaved particles (Table 2). The highest $NH_4^+$ accumulation is found in samples with the largest difference in N/C ratios between autoclaved and non-autoclaved particles (Table 2, 904-20m, 898-100 m)."**

Line 466: In relation to the 'Unchanged N2O production with higher O2 levels in NO3- treatments...' sentence: Can anoxic niches be ruled out in these experiments? You do note the sampling being during low upwelling and chl period but the settling of small particulates during experiments may create anoxic/low O2 zones to sustain anaerobic processes.

**Anoxic micro niches can never be fully ruled out, if not investigated. The Chlorophyll concentrations were in deed low for an upwelling area, max 5mg/m3, but on average 1mg/m3 and less. Figure 1, map of the study site was adjusted with Chlorophyll concentrations. The treatment was identical between depth profile samples and manipulation samples, so if the particles settle, they would settle in all of the bottles and create microniches in the samples from the depth profile as well. There is no**

**plausible explanation why more anoxic micro niches should be in the oxygen manipulations compared to the others.**

Sentence line 470-472 Bristow et al 2016 should also be a ref here in relation to kinetics of multi-step processes

> **Bristow et al. 2016a and b were added.**

Line 477: How can you be sure none of the N2O was consumed without further measurements (e.g. 15N-N2)? Production may just be much faster than consumption.

> **We are not able to say anything about N2 production, we can only assume. We added sufficient amounts of 44N2O carrier prior to the incubation to trap 15N-labelled N2O. If N2O reduction is taking place at high rates, we would see a decrease in the N2O pool over time. A plot with the mass 44, 45 and 46 over time was added to the supplements (Figure S2).**

Line 512-515: Confusing sentences, consider rephrasing.

> **Sentence was rephrased (line 565 – 567): " While high N$_2$O yields are usually found in low O$_2$ waters (<6 µmol L$^{-1}$), in this study AO had also high yields at higher oxygen concentrations, 0.9 % at 30 µmol L$^{-1}$ O$_2$ compared to previous studies (0.06% at > 50 µmol L$^{-1}$ Ji et al. 2018a)."**

Line 521: This is a bit of an oversimplification - because something is below detection doesn't necessarily mean nothing is happening, more likely a tight coupling between consumption and production (e.g. see Figure 4 in Klawonn et al 2019 and Figure 3 in Olofsson et al 2019 references). Could there be a dilution of your 15NH4+ pool to consider due to rapid cryptic cycling on shorter scales than your experiments? Ideally 15NH4+ and total NH4+ would be followed through the time series to check for dilution effects. Both show very rapid NH4+ turnover (within ~5h) in oligotrophic waters

> **The 15NH4+ substrate was not measured on the GC-IRMS because high $^{15}$N label/ almost pure tracer is always problematic to analyze. We added the possibility of an overestimation of hybrid production to the method section line 217 and rephrased the wording here to (line 573-575) "Even though, *in situ* NH$_4^+$ is below detection in almost all water depths (f > 0.9), there remains the potential for $^{15}$NH$_4^+$ pool dilution by remineralization and DNRA during during the incubation. Despite below detection limit studies have shown fast turnover for NH$_4^+$ (Klawonn et al. 2019)."**

Line 532: If measured, the accumulation and consumption of intermediates (e.g. NO2-) could also be used to imply biotic vs abiotic mechanisms (e.g. Betlach and Tiedje 1981 reference).

> **We measured NO2- concentrations and isotopic composition in the 15NH4+ treatments, but not other intermediates like NH2OH or NO. The change in concentration was below our detection limit 50nM. Abiotic N2O production was seen in the 15NO2- treatments in the anoxic depth. A supplementary figure was added Figure S9.**

Line 560-3: Could a 15N recovery/inventory be calculated for the experiments (e.g. 15N recovery from initial substrate, measured intermediates and 15N-N2O?) This could help infer a % N2O production from denitrification which is important for putting the N2O production from denit in context – i.e. how do variations in O2 impact the proportion of N2O produced by denit relative to N2?

**The reviewers make a good point, having both the N2 and N2O production from the same flask at low rates would be very nice. We do not think that there is a way we can come to a N2O /N2 yield without measuring $N_2$. The biological variations in the NO3- pool were so big that the little change in 15NO3- was too small to be detected. Therefore, the yield of N2O/NO2- was calculated.**

Fig 4 b, c & Fig 6 b: consider zoomed-in insert of x-axis (e.g. similar to Fig S5)

**Zoom ups are added into the figures.**

Figure S5: Seems to be two different slopes here from manipulated vs natural O2 gradients – could also be discussed in relation to purging artefacts.

**Both slopes are indicated in figure S7 now. Yes, this could be a purging artefact, but the scatter at the lower range is very high.**

---

## Author Comment (AC3) · 31 Jan 2020

Dear referee, we appreciate the suggestion of the paper by Torres-Beltrán et al., 2019 whom did a great job in comparing in situ sampling and conventional Niskin/Go-flow bottle sampling. We refer to the paper in our manuscript. Best, Frey et al.

---

## Referee Comment (RC4) · Anonymous Referee #2 · 11 Feb 2020

*General comments:*

Can you be sure that there is no DNRA occurring in your experiments – in particular given the Lam et al. 2009 'Revising the N cycle...' paper also off the Peruvian coast. The presence of DNRA would complicate your isotope pairing experiments with 15NO3- and 15NO2- by transferring 15N into the NH4+ pool and you would get 'hybrid'N2O' of 15nh4+ and 15no2- forming 46N2O and be wrongly assigned. DNRA would also potentially dilute your 15NH4+ pool with 14N from background NO3- and alter the assumed 99% labelling in these experiments. I realise the contribution of AO to N2O production is small relative to denit, but the artefacts of DNRA on the rates/data should be discussed as it could lead to some N2O from AO being 'hidden'.

**The reviewer raises a very important point and no, we cannot be sure that the occurrence of DNRA is impacting our results. We added this consideration to the manuscript in line 216 - 232. "Nevertheless, this assumption brings some initial considerations which need to be accounted for. There is a potential for overestimating hybrid N2O production in 15NO2- incubations by 5% in samples with high NO3- reduction rates. But in incubations from anoxic depths with high NO3- reduction rates, no hybrid N2O production is found at all. For example, accounting for a decrease in fN of the NO3- pool by active NO2- oxidation, the process with highest rates (Sun et al. 2017), had an effect of only ± 0.2 % on the final rate estimate. The presence of DNRA complicates 15N-labelling incubations because it can change f in all three tracer experiments. In the vicinity of DNRA in 15NO3- incubations, 15NO2- and 15NH4+ can be produced from 15NO3- which can contribute to 46N2O production by AO. Even when a maximum DNRA rate (20 nM d-1 in Lam et al. 2009) is assumed to produce 0.02 nM 15NH4+ in 24 h with all of it being oxidized to N2O (max. N2O production from AO 0.16 nM d-1, this study), its contribution to 46N2O production is likely minor and within the standard error of N2O production rates from NO3-. Hence an overestimation of the N2O production rates is unlikely. The same applies in incubations with 15N-NO2- when DNRA produces 15NH4+, additional 46N2O can be produced with a hybrid mechanism by AO not accounted for in the present rate calculations. In 15NO2- incubations with high starting f (>0.7) the production of 14NO2- by NO3- reduction (which decreases f) leads to an underestimation by max. 9%, whereas in incubations with a low f (<0.3) the effect is less with max. 3 % underestimation of N2O production rates. In 15NH4+ incubations (f>0.9), max. DNRA rate would lead to an underestimation of 3.5 %. "**

**Specific comments:**

Section 2.1: As with other papers with many sites, sampling points and manipulation experiments a written methods text quickly becomes very complicated with different additions, concentrations, replicates, time points etc. I think as a result of the text being quite confusing some information has been missed/is unclear. Adding a table of experiments, stations, variables, sampling routine (e.g. time points), number of replicates, other factors (e.g. whether O2 was measured in vials) would be informative/helpful to readers who are interested in comparing/replicating experiments.

**It is correct, that such set ups can get confusing very quickly, but in table 1 stations, depths, measured variables and the kind of experiment performed are given. However, we added one column with the kind of tracer addition we did. The replicates and time points did not vary between experiments and hence is only stated in the test. We only measured oxygen in one bottle with each incubation per depth or treatment, which was also consistent and written in the text line 168/169.**

My apologies – I missed this table but think the added information now makes it clearer.

Line 145 (O2 manipulation experiments): Why was such a 'coarse' O2 range used compared to previous studies which use O2 manipulations generally below 1-2µM (e.g. Dalsgaard et al 2014, Bristow et al 2016)?

**In Bristow et al. 2016 a and b the maximal oxygen concentration in their manipulation experiments was 10uM and 20uM dissolved oxygen, so we are not quite sure what the referee means. Dalsgaard et al 2014 performed a really nice microcosm experiment, where oxygen concentrations were monitored online in the flask they subsampled. In our case, each time point was a separate bottle making it impossible to use such an approach. For the experimental design in this study, it was important to choose oxygen levels where we can be sure that oxygen concentrations are different enough from each other that we can differentiate the two treatments (f.e. 100nM and 200nM would be tricky to tell apart with our standard deviations of 180nM and 240nM over 24h). We added a plot of oxygen over time into the supplements Figure S1.**

I meant that in Bristow et al. and Dalsgaard et al that a lot of their measurements are concentrated below 1-2 uM oxygen and fewer concentrations in the 'higher' 10-20uM range… i.e. focusing on the concentrations where the inhibition/regulation really 'happens'. But I understand the reasons you describe above given the standard deviations of O2 measurements and without more sensitive sensors it would be difficult to designate concentrations, I agree. I appreciate that Dalsgaard et al do have a nice reactor/microcosm set up which I realise is very specialised for precisely these experiments and with larger volumes than the serum vials – also that it is a lot of work with these types of experiments. I think it would still be good to add a sentence/statement as to why 'your' oxygen concentrations were chosen (e.g. given the reasons above, SD in measurements etc) if possible.

Line 466: In relation to the 'Unchanged N2O production with higher O2 levels in NO3- treatments...' sentence: Can anoxic niches be ruled out in these experiments? You do note the sampling being during low upwelling and chl period but the settling of small particulates during experiments may create anoxic/low O2 zones to sustain anaerobic processes.

**Anoxic micro niches can never be fully ruled out, if not investigated. The Chlorophyll concentrations were in deed low for an upwelling area, max 5mg/m3, but on average 1mg/m3 and less. Figure 1, map of the study site was adjusted with Chlorophyll concentrations. The treatment was identical between depth profile samples and manipulation samples, so if the particles settle, they would settle in all of the bottles and create microniches in the samples from the depth profile as well. There is no plausible explanation why more anoxic micro niches should be in the oxygen manipulations compared to the others.**

But if there is the same amount of particles in all vials/O2 manipulations then there is potential for some anoxic processes to be 'unaffected' by O2 additions - with some changes in anoxic microsite volume with O2 diffusion into particles. I realise this is hard to rule out – especially as you collect small particles from the water column to use, indicating that they are there. I think it would be important to write something shortly about why you consider it unlikely that any (significant) anoxic niches occur.

Line 477: How can you be sure none of the N2O was consumed without further measurements (e.g. 15N-N2)? Production may just be much faster than consumption.

**We are not able to say anything about N2 production, we can only assume. We added sufficient amounts of 44N2O carrier prior to the incubation to trap 15N-labelled N2O. If N2O reduction is taking place at high rates, we would see a decrease in the N2O pool over time. A plot with the mass 44, 45 and 46 over time was added to the supplements (Figure S2).**

Shortly suggest/indicate benefits of also measuring other end products (e.g. 15N-N2 and maybe also 15NH4+ from DNRA) in the text (i.e. how does the 'efficiency' of denit change with changing O2)

Line 521: This is a bit of an oversimplification - because something is below detection doesn't necessarily mean nothing is happening, more likely a tight coupling between consumption and production (e.g. see Figure 4 in Klawonn et al 2019 and Figure 3 in Olofsson et al 2019 references). Could there be a dilution of your 15NH4+ pool to consider due to rapid cryptic cycling on shorter scales than your experiments? Ideally 15NH4+ and total NH4+ would be followed through the time series to check for dilution effects. Both show very rapid NH4+ turnover (within ~5h) in oligotrophic waters

**The 15NH4+ substrate was not measured on the GC-IRMS because high 15N label/ almost pure tracer is always problematic to analyze. We added the possibility of an overestimation of hybrid production to the method section line 217 and rephrased the wording here to (line 573-575) "Even though, *in situ* NH4+ is below detection in almost all water depths (f > 0.9), there remains the potential for 15NH4+ pool dilution by remineralization and/or DNRA during  the incubation. Despite   studies have shown fast turnover for NH4+ (Klawonn et al. 2019).... However/thus"**

---

## Author Comment (AC5) · 18 Feb 2020

Response to second round of Referee #2 suggestions:

Dear Referee,
we appreciate your additional suggestions to our first response and integrated them into our current manuscript. We outlined our point to point reply below (in bold).

General comments:
Thanks, I realise it will be a small % but important to acknowledge. It would also be good to add a sentence in the discussion to suggest that DNRA(& anammox) are measured in addition in future work to rule out potential artefacts – there are now several (sediment) papers on the artefacts of the coocurrence of NO3- reducing processes on the IPT assumptions.

**We added the importance of measuring N2 production in future studies in the discussion:**
**Line 574: "In future $^{15}$N -labelling studies, DNRA should be measured to rule out potential pool dilution by the co-occurrence of $NH_4^+$ production. "**
**Line 482- 485: "$N_2$ production measurements (from anammox and denitrification) were not performed in this study, but should be carried out in future studies to account for potential artefacts by co-occurring $NO_3^-$ reduction processes."**

I think you just need to change "In the vicinity of DNRA in 15NO3- incubations..." to "In relation to DNRA in 15NO3- incubations..."

**We rephrased the whole sentence in the results section to: "In $^{15}NO_3^-$ incubations, active DNRA produces $^{15}NO_2^-$ and $^{15}NH_4^+$ from $^{15}NO_3^-$ which can contribute to $^{46}N_2O$ production by AO."**

I meant that in Bristow et al. and Dalsgaard et al that a lot of their measurements are concentratedbelow 1-2 uM oxygen and fewer concentrations in the 'higher' 10-20uM range... i.e. focusing on theconcentrations where the inhibition/regulation really 'happens'. But I understand the reasons youdescribe above given the standard deviations of O2 measurements and without more sensitive sensorsit would be difficult to designate concentrations, I agree. I appreciate that Dalsgaard et al do have anice reactor/microcosm set up which I realise is very specialised for precisely these experiments and with larger volumes than the serum vials – also that it is a lot of work with these types of experiments. I think it would still be good to add a sentence/statement as to why 'your' oxygen concentrations werechosen (e.g. given the reasons above, SD in measurements etc) if possible.

**We think that the standard deviations of the different oxygen levels explain why we did not resolve the lower end better and did not add anything there. However, we agree with the referee that it does not become clear why a larger range was applied for the 15NO3- treatments, so we explained that better:**
**Line 167-169: "For the $^{15}$N-$NO_3^-$ incubations two more $O_2$ treatments with 21.5 ± 2.8 and 30.2 ± 3.35 µM $O_2$ were carried out to extend the range of a previous study in which $N_2O$ production from $^{15}NO_3^-$ did not decrease up to $O_2$ concentration of 7 µM (Ji et al. 2018)."**

But if there is the same amount of particles in all vials/O2 manipulations then there is potential for some anoxic processes to be 'unaffected' by O2 additions - with some changes in anoxic microsite volume with O2 diffusion into particles. I realise this is hard to rule out – especially as you collect small particles from the water column to use, indicating that they are there. I think it would be important to write something shortly about why you consider it unlikely that any (significant) anoxic niches occur.

**We do not consider it unlikely that anoxic niches occur, but we do think that anoxic niches do not explain the large difference in response of N2O production at high oxygen levels in the depth profiles (no to little N2O production) compared to the manipulated oxygen treatments (very high N2O production), because the potential for anaerobic microsites is given in all incubations. We added the potential for anaerobic processes inside microniches in line 530 – 534: „ It further indicates that high $N_2O$ production from $NO_3^-$ in high oxygen treatments is unlikely an effect of anoxic micro niches. While anoxic micro niches in batch incubations can never be fully ruled out, there is no reason why they should systematically change N2O production in $NO_3^-$ from $NO_2^-$ incubations at the same oxygen treatment. "**

Shortly suggest/indicate benefits of also measuring other end products (e.g. 15N-N2 and maybe also 15NH4+ from DNRA) in the text (i.e. how does the 'efficiency' of denit change with changing O2)

**We added the advantage of measuring several potential end products in line 574 about DNRA and in line 482- 485 about anammox and denitrification (see first comment). The advantage of having production rates of N2O and N2 together is already discussed starting in line 485, and also starting in line 532, where we highlight the value of having the N2O yields. The different responses/efficiency of denitrification to oxygen is extensively discussed in lines 523 onwards.**

Some kind of 'conclusion' is needed at the end of the last sentence in relation to your study. Papers referring to 'cryptic' biogeochemical cycling in ODZ waters would also be nice to include in relating to 'hidden' processes.

**As suggested, we added a conclusion to in line 581: "Even if hybrid $N_2O$ production rates are overestimated, it remains the major $N_2O$ production mechanisms of AO in this study."**

**In this paragraph we want to explain the occurrence of hybrid N2O formation rather than hidden process – so we did not add papers on cryptic cycling there.**